# Longitudinal functional imaging of VIP interneurons reveals sup-population specific effects of stroke that are rescued with chemogenetic therapy

Mohamad Motaharinia[1,4], Kim Gerrow[1,4], Roobina Boghozian[1], Emily White[1], Sun-Eui Choi[1], Kerry R. Delaney[2] & Craig E. Brown [1,2,3 ✉]

Stroke profoundly disrupts cortical excitability which impedes recovery, but how it affects the function of specific inhibitory interneurons, or subpopulations therein, is poorly understood. Interneurons expressing vasoactive intestinal peptide (VIP) represent an intriguing stroke target because they can regulate cortical excitability through disinhibition. Here we chemogenetically augmented VIP interneuron excitability in a murine model of photothrombotic stroke and show that it enhances somatosensory responses and improves recovery of paw function. Using longitudinal calcium imaging, we discovered that stroke primarily disrupts the fidelity (fraction of responsive trials) and predictability of sensory responses within a subset of highly active VIP neurons. Partial recovery of responses occurred largely within these active neurons and was not accompanied by the recruitment of minimally active neurons. Importantly, chemogenetic stimulation preserved sensory response fidelity and predictability in highly active neurons. These findings provide a new depth of understanding into how stroke and prospective therapies (chemogenetics), can influence subpopulations of inhibitory interneurons.

[1] Division of Medical Sciences, University of Victoria, Victoria, BC V8P 5C2, Canada. [2] Department of Biology, University of Victoria, Victoria, BC, Canada. [3] Department of Psychiatry, University of British Columbia, Vancouver, BC, Canada. [4]These authors contributed equally: Mohamad Motaharinia, Kim Gerrow. ✉email: brownc@uvic.ca

Stroke leads to impairments in sensory, motor, and cognitive function that disrupt everyday activities and independent living. Although some patients will experience partial recovery in the weeks to months that follow a stroke, the majority will be left with permanent disability[1,2]. One plausible explanation for limited recovery is that surviving, functionally related neural circuits undergo persistent disturbances in synaptic structure and neuronal excitability. In support of this, human and experimental animal studies have shown the stroke leads to synapse loss on excitatory neurons and connections, diminished sensory responses and abnormal electrical or metabolic brain activity patterns[3–8]. Regions surrounding the stroke (e.g. peri-infarct cortex) are particularly susceptible to ischemia induced dysfunction. However, they also appear to play a critical role in recovery[9,10] given that the re-emergence of normal activity patterns in peri-infarct cortex correlate with improvements in limb use/abilities[11], whereas disruption of this region can re-instate functional deficits[12–15]. While considerable progress has been made in documenting stroke related changes in brain structure and function, the majority of studies have focused on excitatory circuits and relied on widefield/mesoscopic or single time-point imaging approaches. Therefore we still lack a cell specific appreciation of how stroke affects the function of neurons over time, especially within inhibitory interneurons or even within subsets of a specific interneuron class[16,17].

Seminal studies from the Carmichael lab revealed that increased GABAergic inhibition could play a major role in limiting the return of cortical excitability and sensorimotor function after stroke[18]. Although changes in post-stroke inhibition can be partially explained by alterations in GABA transporter proteins or extra-synaptic GABA_A receptors[19], it is also very likely that stroke disrupts the wiring diagram of inhibitory circuits[9,20,21]. Cortical excitability is largely governed by the interplay between excitatory neurons and inhibitory interneurons that express either parvalbumin, somatostatin, or vasoactive intestinal peptide (PV, SOM, and VIP, respectively). In general, PV and SOM interneurons provide direct inhibition onto excitatory neurons near the soma or distal dendrites, respectively[16]. However, the activity of PV and SOM interneurons can be regulated by inhibitory synaptic connections from VIP neurons[22–24]. Given that cortical VIP interneurons receive excitatory inputs from sensory nuclei in the thalamus and local cortical neurons[25], the net effect of activating these disinhibitory VIP circuits is to enhance cortical responses to a sensory stimulus[26–28]. Not surprisingly, VIP neurons have been implicated in regulating developmental, learning, and experience dependent plasticity of visual cortical circuits[29,30]. Whether these disinhibitory circuits are disrupted by stroke or could be targeted with a therapy to restore cortical excitability and function, has not been explored.

In the present work, we hypothesized that chronic stimulation of VIP interneurons could help restore cortical excitability after stroke and promote the recovery of forepaw sensorimotor function. We opted for a Designer Receptors Exclusively Activated by Designer Drugs (DREADD-hM3Dq) chemogenetic approach[31] because it would allow us to enhance VIP interneuron excitability across the depth of the cortex for at least an hour each day after a single injection of the hM3Dq receptor ligand, Clozapine N-oxide (CNO). Using a battery of behavioral, electrophysiological and imaging approaches, we show that chemogenetic stimulation of VIP interneurons enhances sensory responses in peri-infarct cortex and improves recovery of sensori-motor abilities. Of note, we found that the disruptive effects of stroke and benefits of chemogenetic therapy were mostly experienced within a subset of highly active VIP interneurons. Our findings provide a framework for understanding the effects of stroke at a cell and subpopulation specific level, as well as support the concept that augmenting cortical excitability through disinhibitory interneurons can facilitate stroke recovery.

## Results

### Chemogenetic stimulation of VIP interneurons enhances somatosensory cortical responses

A number of previous studies from our lab and others has shown that focal cortical stroke leads to a long-lasting dampening of cortical excitability and responsiveness to sensory stimulation that correlates with deficits in sensorimotor function of the affected limb[7,8,32–34]. Based on the rationale that VIP interneurons could act as a disinhibitory circuit and thus potentially restore cortical responsiveness after stroke, we first characterized the effects of excitatory hM3Dq stimulation on VIP neurons in mouse somatosensory cortex. As summarized in Fig. 1a, adult male VIP-cre mice were first microinjected with AAV (AAV2.hSyn.DIO.hM3Dq.mcherry; two injections of 0.5 μL $1 \times 10^{12}$ vg/mL spaced 1 mm apart, +1.5 and 2.5 mm lateral of bregma) to drive hM3Dq expression in VIP neurons in the right forelimb primary sensorimotor cortex (FLS1). Controls consisted of mice that were injected with AAV1.flex.eGFP or sham surgery. Since VIP-cre expression is highly sensitive and specific for labeling all VIP cortical interneurons[35], a subset of mice were injected with both AAVs (AAV1.flex.eGFP and AAV2.hSyn.-DIO.hM3D(Gq).mcherry) in order to confirm robust hM3Dq expression in cortical VIP neurons (Fig. 1b). Three weeks after AAV injections, mice were subjected to photothrombotic or sham control stroke in the right forelimb somatosensory cortex. Stroke-related damage did not encompass the entire FLS1 cortex and typically spared a small portion of medial or posterior FLS1 cortex. One week after stroke or sham control, mice were lightly anesthetized with urethane and forepaw-evoked field potentials were recorded in layer 2/3 of peri-infarct or uninjured cortex. In both stroke and sham control mice, injection of the hM3Dq agonist CNO (0.3–0.5 mg/kg, i.p.) significantly increased the amplitude of sensory-evoked cortical field potentials in mice that expressed the excitatory DREADD hM3Dq relative to baseline or vehicle injection (Fig. 1c, d and Supp. Fig. 1a, b). There was no significant difference in peak response amplitude when comparing 0.3 mg/kg vs 0.5 mg/kg doses of CNO (unpaired t-test, $t_{(3)} = 1.21$, $p = 0.31$). The effect of CNO on cortical responses peaked around 30 min and remained above baseline levels for up to 90 min after injection (Fig. 1e and Supp. Fig. 1c). Importantly, CNO injection had no detectable effect on response amplitude if mice did not express hM3Dq (Fig. 1d and Supp. Fig. 1a), thereby minimizing the possibility that CNO, by itself, could induce changes in cortical excitability. To determine if chemogenetic stimulation was sufficient to alter regional cerebral blood flow, we performed Laser Doppler flowmetry in stroke and sham control mice. Our experiments revealed that CNO injection, regardless of whether mice expressed hM3Dq or were subjected to stroke, had no effect on basal cerebral blood flow (Supp. Fig. 1d, e). Since we could not detect any changes in blood flow, we conducted a positive control experiment (exposure to 6% $CO_2$) and as expected, this resulted in a significant increase in cerebral blood flow (Supp. Fig. 1d, e). These findings suggest that our paradigm for DREADD based stimulation of VIP neurons was sufficient to increase cortical sensory responses but insufficient to significantly alter regional blood flow.

### Chronic chemogenetic stimulation after stroke improves functional recovery

Since chemogenetic stimulation of VIP neurons could enhance cortical responses to forepaw stimulation after stroke, we next assessed whether this approach could facilitate the recovery of sensori-motor paw function. As shown in Fig. 1a, adult male VIP-cre mice were injected with AAVs to drive

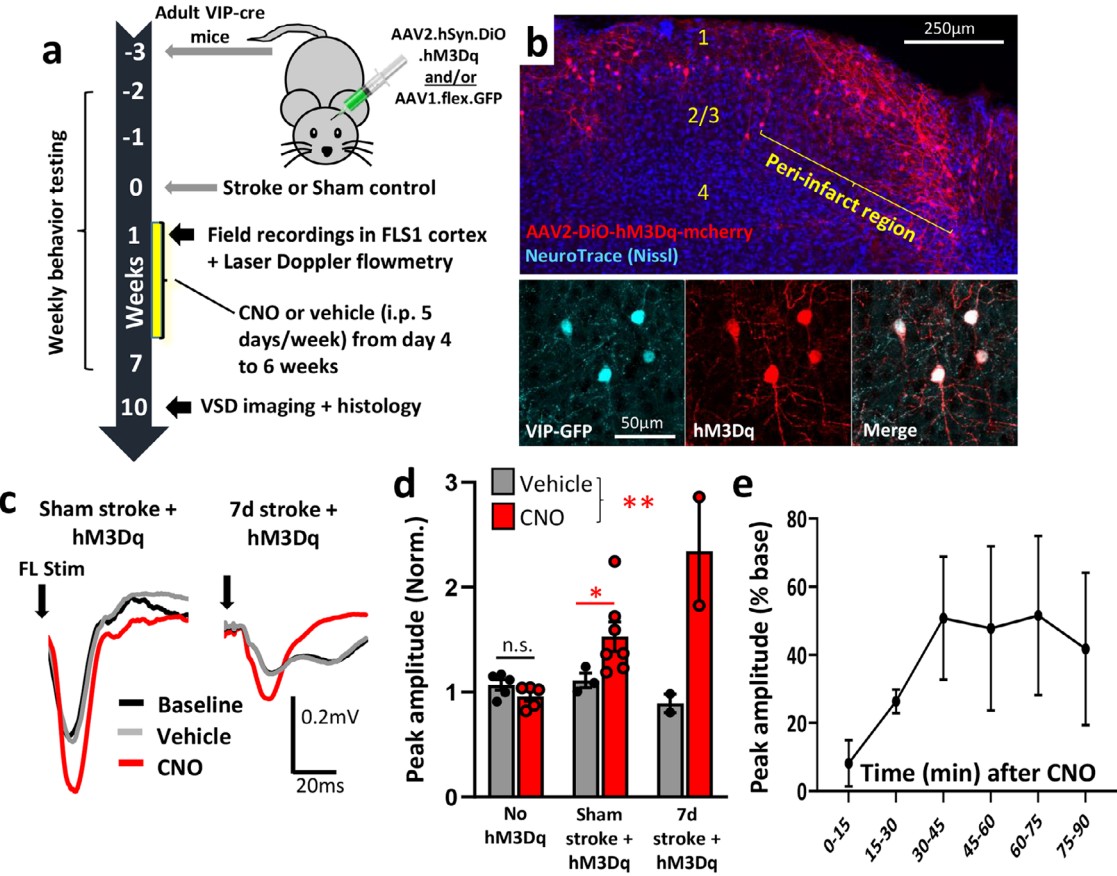

**Fig. 1 Chemogenetic stimulation of VIP interneurons enhances cortical responses to forepaw touch. a** Diagram summarizing the timeline for surgical procedures (AAV injection, stroke/sham surgery) and assessment of cortical excitability with field recordings, regional blood flow and behavior in mice that received chemogenetic or control stimulation. **b** Confocal images showing the distribution of DREADD hM3Dq expressing neurons in the cortex after stroke (top; images representative of all three experimental mice). Bottom row, confocal images showing that hM3Dq expression was limited to VIP neurons expressing cre-recombinase. **c** Representative traces of forelimb-evoked (one 5 ms deflection) field potentials in somatosensory cortex of sham stroke controls (left) or peri-infarct cortex 7 days after stroke (right). Mice expressing hM3Dq were recorded at baseline, and then injected with vehicle followed by CNO (0.3 mg/kg). Traces represent an average of 45 stimulation sweeps, collected ~30 min after injection. **d** Graph shows the peak amplitude of forelimb-evoked responses (normalized to baseline) in each experimental group approximately 30 min after CNO injection (0.3 or 0.5 mg/kg). In all mice expressing hM3Dq, CNO significantly increased response amplitudes compared to vehicle ($n = 9$ mice; two-way ANOVA for CNO Treatment: $F_{(1,10)} = 18.52$, **$p = 0.002$; Sham stroke CNO vs vehicle: unpaired two-sided t-test $t_{(8)} = 2.68$, *$p = 0.013$). By contrast, CNO had no effect if mice did not express hM3Dq ($n = 5$ mice without hM3Dq; unpaired two-sided $t$-test, $t_{(8)} = 1.61$, $p = 0.14$). **e** Time-dependent increase in amplitudes of forelimb-evoked potentials recorded in layer 2/3 of the somatosensory cortex in mice expressing hM3Dq following CNO injection ($n = 3$ mice; 0.3 mg/kg, i.p.). n.s. not significant. Data show means ± S.E.M.

hM3Dq expression in VIP neurons or control injection of AAV1.flex.eGFP. Mice were then randomly assigned to receive chemogenetic therapy that consisted of daily injections of CNO (0.3 or 0.5 mg/kg, i.p.) starting 4 days after stroke and continuing for up to 6 weeks (1 injection per day/5 days per week). Therapy commenced at 4 days after stroke to avoid over-stimulating the cortex in the acute phase, which could exacerbate ischemic damage. The two control groups consisted of: (a) mice expressing hM3Dq but injected with vehicle or (b) mice that did not express hM3DGq (e.g. received AAV1.flex.eGFP) but received chronic CNO injections. Based on our previous studies, we used the horizontal ladder walking and the adhesive tape removal test to assess forepaw function after stroke[36,37]. Of note, behavioral tests were always conducted before daily CNO/vehicle injections (~23 h after last injection), to minimize the possibility of any acute effects of CNO treatment. For the ladder test, stroke led to a significant reduction in the percentage of correct forepaw placements in both groups at 1 day and 1 week after stroke (Fig. 2a left panel). However relative to the control group, mice that received chemogenetic stimulation began to show superior performance in

the ladder test at 2 weeks recovery (Fig. 2a). This improved performance persisted throughout the remainder of the experiment, including week 7 which was 1 week after CNO treatment had ceased. Importantly, enhanced recovery in mice that received chemogenetic stimulation was observed when running both types of control groups, as well as when run by two different experimenter cohorts whom were blind to treatment. In order to determine if the benefits of the chemogenetic treatment were affected by sex, we compared the difference in correct paw placements in treated mice verses controls. Our analysis indicated that treatment effects were not influenced by sex given that improved ladder performance was similar for males and females (two-way ANOVA; Main effect of Sex: $F_{(1,72)} = 1.31$, $p = 0.26$). In contrast to the ladder test, the adhesive tape test was not sensitive in detecting lasting impairments in paw function beyond the first day after stroke (Supp. Fig. 2a, b; $p > 0.05$ for all time points after day 1). This lack of sensitivity was surprising to us as it diverges from our previous stroke studies using C57BL6 mice, suggesting that mouse strain could influence behavioral assessment of stroke recovery. In combination, these results indicate that chemogenetic

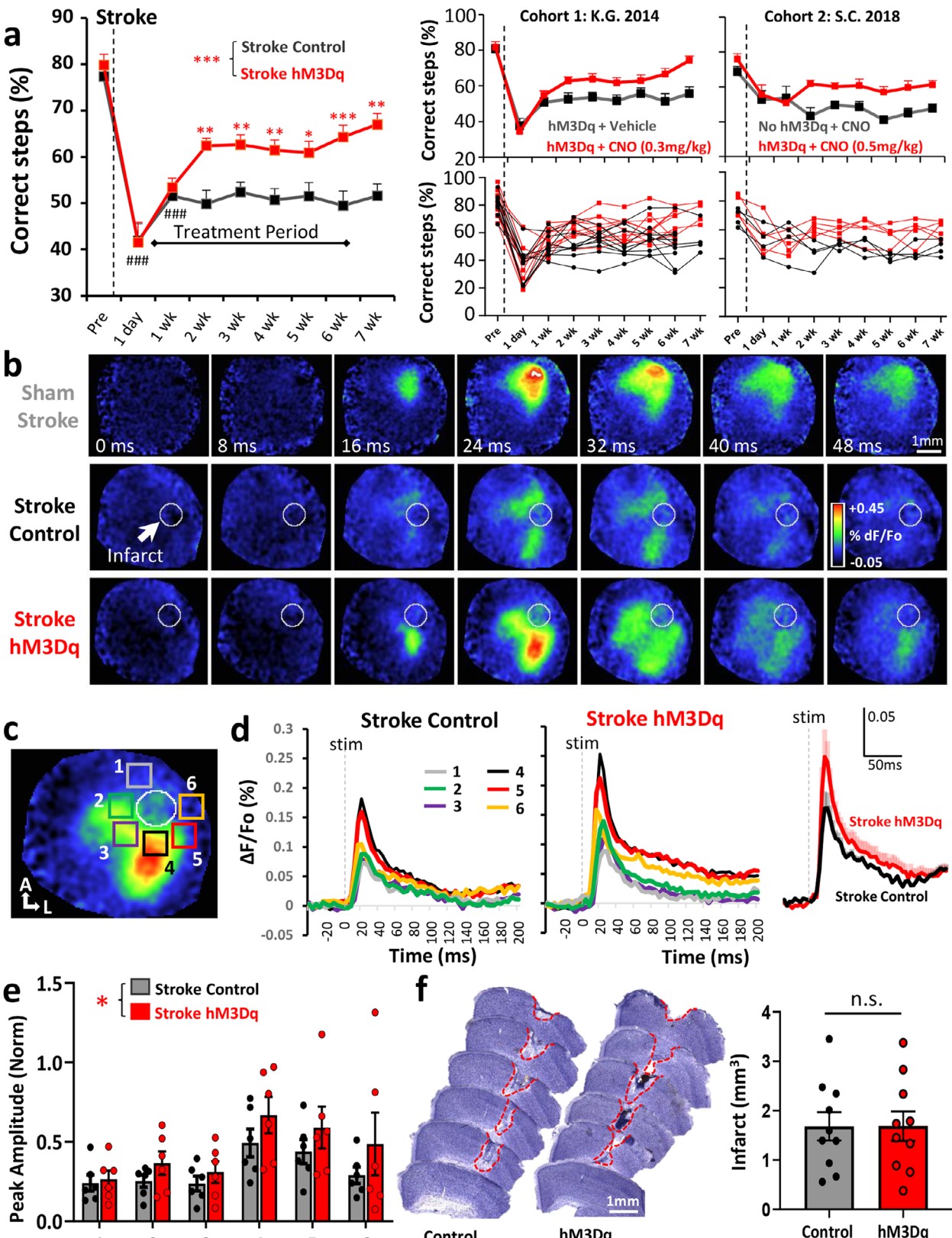

stimulation of VIP neurons improves recovery of paw function in behavioral tests that are sensitive in detecting long-lasting impairments.

Next we examined whether improved sensorimotor forepaw function was associated with changes in forelimb-evoked cortical responses by 10 weeks recovery. Consistent with previous voltage

sensitive dye (VSD) imaging studies[32,38], vibro-tactile stimulation of the left forepaw normally leads to a robust depolarization in the right forelimb somatosensory cortex (Fig. 2b). In stroke affected mice that received chemogenetic stimulation ("Stroke +hM3Dq"), forepaw-evoked depolarizations in peri-infarct cortex were larger in amplitude than those receiving the control

**Fig. 2 Chronic chemogenetic stimulation improves recovery of forelimb sensori-motor function and cortical responsiveness after stroke. a** Left: graph shows the average number of correct steps (% of total steps) per time point on the horizontal ladder for mice that received control treatment ($n = 14$ mice) or chronic chemogenetic treatment ($n = 15$ mice) from day 4 to 6 weeks after stroke. Chemogenetic treatment significantly increased the % correct steps on the horizontal ladder test compared to control treatment (two-way ANOVA, main effect of treatment: $F_{(1,231)} = 42.4$, $p < 0.0001$). Right: graphs show 2 cohorts of behavioral studies from two different blinded observers (K.G. in 2014 and S.C. in 2018). All mice in Cohort 1 were male whereas all mice in Cohort 2 were female. These graphs show the benefits of chemogenetic stimulation relative to two types of stroke controls: (a) mice expressing hM3Dq that received vehicle injection or (b) mice that did not express hM3Dq but received CNO injections. **b** VSD image montages showing forepaw-evoked depolarizations in the right somatosensory cortex in sham controls or stroke affected mice that received control or chemogenetic stimulation (images representative of mean responses observed in each experimental group). Stroke control mice consisted of those that expressed hM3Dq but received vehicle injections. **c** Image delineates where VSD ΔF/Fo measurements were sampled relative to the infarct. **d** Graphs show mean forelimb-evoked depolarization in each peri-infarct ROI (#1-6) in stroke affected mice that received control or chemogenetic stimulation treatment ($n = 6$ mice/group). Right: traces show the average forelimb-evoked response (shaded area represents SEM) for all ROIs and all mice within each group. **e** Bar graph shows that chemogenetic treatment significantly increased the peak amplitude of forelimb responses (normalized to HL responses) in peri-infarct cortex (two-way ANOVA, Main effect of Treatment: $F_{(1,60)} = 5.12$, $p = 0.027$). **f** Representative cresyl violet staining of coronal sections illustrating the location and extent of cerebral infarcts. Infarct volume did not differ between groups ($n = 10$ mice/group; unpaired two-sided t-test, $t_{(18)} = 0.12$, $p = 0.99$). ###$p < 0.001$ for t-test comparisons against pre-stroke. *$p < 0.05$, **$p < 0.01$, ***$p < 0.001$ for t-test comparisons between Stroke hM3Dq vs Stroke Control. n.s. = not significant. Data show means ± S.E.M.

treatment (Fig. 2b–d). Normalizing these FL evoked responses (to HL responses) to control for between animal differences in overall excitability levels that are inherent with VSD imaging experiments, revealed significantly larger response amplitudes in mice that received chemogenetic treatment versus controls (Fig. 2e). There were no significant group differences in the time to peak or half-width of forelimb-evoked responses (two-way ANOVA, Main effect of Treatment on time to peak: $F_{(1,60)} = 1.59$, $p = 0.21$; Main effect of Treatment on half-width: $F_{(1,60)} = 1.24$, $p = 0.27$). It is important to note that the volume of cerebral infarcts between the stimulated and controls groups were virtually identical (Fig. 2f). These results show that chemogenetic stimulation after stroke leads to a long-term enhancement of cortical responsiveness to forepaw touch which cannot be explained by differences in infarct volume.

**Chemogenetic therapy mitigates the loss of sensory responses in VIP neurons after stroke.** In order to provide a cellular understanding of how chemogenetic stimulation could influence stroke recovery, we employed two-photon microscopy to image forepaw-evoked calcium responses in VIP neurons before and after stroke. To do this, adult male VIP-cre mice were micro-injected with AAVs (AAV1.synapsin.Flex.GCaMP6s and AAV2.hSyn.DIO.hM3D(Gq).mcherry) to drive GCaMP6s and hM3Dq expression in VIP neurons in the FLS1 cortex (Fig. 3a). Following installation of a cranial window, we used intrinsic optical signal (IOS) imaging to map the FLS1 cortex which allowed as to target 2-photon imaging of VIP neurons in regions near the posterior-medial aspect of the FLS1 border (see pink box in Fig. 3a), which our previous studies have shown is where forelimb evoked responses usually re-emerge after stroke. Mice were then randomly selected to receive sham or photothrombotic stroke targeted to FLS1 cortex adjacent to where VIP neurons were imaged. Mice were then randomly assigned to receive daily CNO or vehicle injections (0.5 mg/kg per day i.p., 5 days a week) starting at day 4 post-stroke and continuing up to 4 weeks recovery (Fig. 3a). Mice that received chemogenetic therapy ("Stroke hM3Dq") were compared to a sham stroke ("Sham stroke") and a stroke group that received vehicle injections ("Stroke Control"). We did not run an additional CNO treated control group (e.g. CNO injections in mice that do not express hM3Dq) given our previous experiments showing that CNO injection, in the absence of hM3Dq, has no detectable effect on VIP neuron excitability or recovery from stroke (see Figs. 1d and 2a). First, we confirmed an abundance of VIP neurons in superficial FLS1 cortex that expressed GCaMP6s (Fig. 3b, top row). Moreover, using post-

mortem immunohistochemistry we show that most GCaMP6s-expressing VIP neurons imaged in vivo, also expressed hM3Dq (Fig. 3b, bottom row). GCaMP6s expression was stable over many weeks which allowed us to repeatedly image forepaw-evoked calcium responses in the same VIP neurons before and after targeted stroke to FLS1 cortex (Fig. 3c). In order to assess VIP neuron responses, we imaged somatic calcium transients in response to eight trials of vibro-tactile stimulation (1.5 s duration at a frequency of 100 Hz) of the contralateral forepaw. Mice were imaged under light (1%) isoflurane anesthesia to allow us to stimulate the forepaw in a precise location (dorsal surface of paw) and in a reliable manner to facilitate response comparisons across mice and imaging sessions. It is important to note that mice were always imaged ~23 h after the last vehicle/CNO injection which is well after the time at which the effects of CNO would have worn off (in vivo estimates range from 1 to 10 h)[39]. Before the induction of stroke or sham procedure, forepaw-evoked responses varied between cells and between trials (compare responses in cell 1, 3 and 5 in Fig. 3d). On average, approximately $46.8 \pm 11.0\%$ of all VIP neurons and $45.3 \pm 8.7\%$ of all trials imaged before stroke were classified as responsive to forepaw stimulation. Of note, there were no significant differences between the three experimental groups with respect to the mean % responsive neurons (Supp. Fig. 3a; One-way ANOVA: $F_{(2,14)} = 2.97$, $p = 0.08$), mean % responsive trials (Supp. Fig. 3b; One-way ANOVA: $F_{(2,14)} = 1.93$, $p = 0.09$) or mean peak amplitude of responses (Supp. Fig. 3c; One-way ANOVA: $F_{(2,14)} = 0.44$, $p = 0.65$) at baseline. In addition, we imaged forelimb responses in 3 awake head fixed mice (without stroke) to make sure that sensory responses recorded under isoflurane weren't completely distorted. Although the fraction of responsive cells, trials and peak amplitudes were slightly higher in awake mice, they did not differ significantly from responses measured at baseline under light isoflurane anesthesia (Supp. Fig. 3d). Lastly, to confirm an effect of CNO on VIP neurons in vivo, we imaged forelimb evoked responses in VIP neurons with confirmed hM3Dq expression and found that 0.5 mg/kg CNO significantly increased mean peak amplitude (Wilcoxon paired t-test, $p = 0.018$) and the number of responsive trials (Wilcoxon paired t-test, $p = 0.008$) relative to baseline (Supp. Fig. 1f).

Following the induction of stroke, there was a notable reduction in the responsiveness of VIP neurons in the peri-infarct cortex of control treated mice (Fig. 4a, b). Since the effect of stroke on forepaw responses was strongly related to distance from the infarct border (Fig. 4c, d), we focused our quantitative analysis on neurons within 400 μm of the border. This analysis

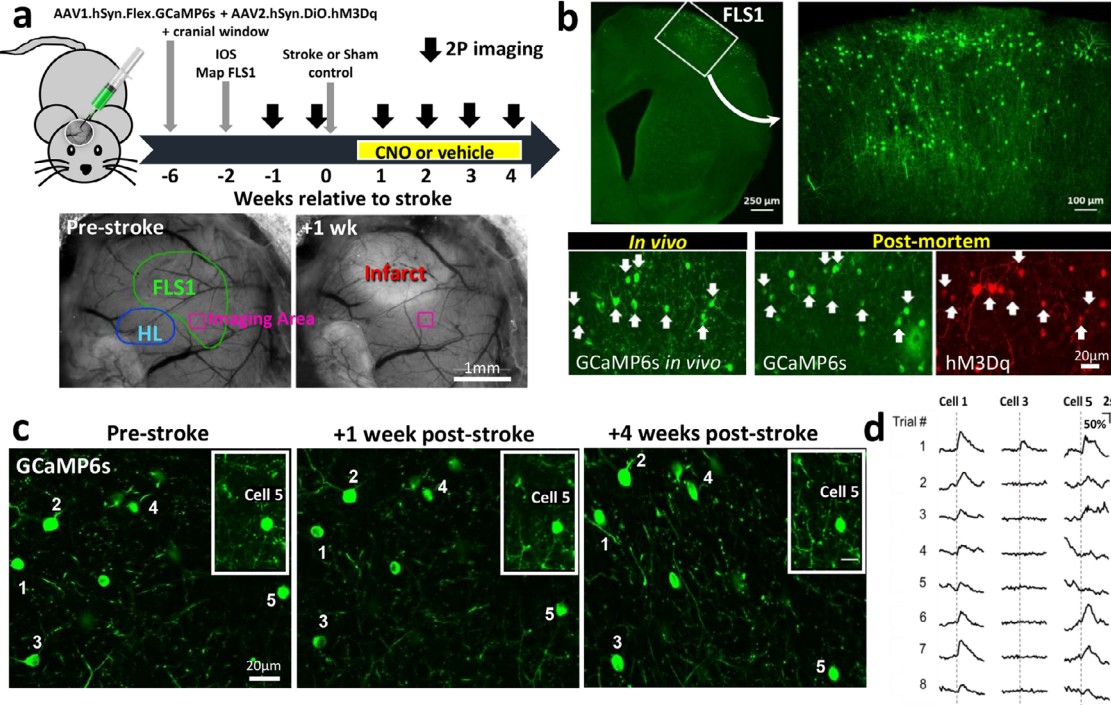

**Fig. 3 Long-term imaging of sensory responses in VIP interneurons. a** Top: Schematic summarizing the timeline of surgical procedures and 2-photon (2P) imaging of VIP interneuron responses to forelimb stimulation. Bottom: surface images show the location of 2P imaging in FLS1 cortex relative to the infarct. The colored contours shown for FLS1 and HLS1 were derived from IOS maps thresholded at 75% peak response amplitude. **b** Top: confocal images show GCaMP6s expression in VIP neurons in forelimb primary somatosensory cortex (FLS1, images representative of all experimental mice). Bottom row: GCaMP6s-expressing VIP neurons imaged in vivo were identified in post-mortem sections and confirmed to express hM3Dq (see white arrows). **c** Calcium imaging of the same VIP interneurons before stroke and afterwards in peri-infarct cortex (images representative of all experimental mice with stroke). **d** Traces show trial by trial variability in forelimb-evoked calcium responses (%ΔF/Fo) in three representative neurons (see numbers in **c**).

revealed that the reduction in sensory responses was most pronounced in the first week after stroke (Fig. 4e) and was manifested by a significant decrease in the: (a) fraction (%) of forelimb responsive neurons ($t_{(42)} = 2.99$, $p < 0.01$), (b) % of responsive trials ($t_{(42)} = 4.11$, $p = 0.02$), and (c) peak amplitude of responses ($t_{(42)} = 2.43$, $p = 0.02$). By contrast, mice that received chemogenetic stimulation therapy did not show the expected reduction in forelimb responsive neurons, trials or peak amplitude in the first week, or any week after stroke (red line in Fig. 4e). It is worth noting that the fraction of responsive neurons and trials were generally stable over the entire imaging period for sham stroke controls (gray line in Fig. 4e), thereby ruling out the possibility that degradation of imaging conditions over time could explain these stroke related effects on VIP neuron responsiveness. We also did not find a significant group difference in "resting" somatic calcium fluorescence (Fo) over time (Fig. 4f; two-way ANOVA: $p > 0.05$ for Main effects of Group, Time or Interaction). Furthermore, we plotted the peak amplitude and the number of responsive trials for each neuron as a function of raw resting somatic calcium levels (Fig. 4g, h, respectively) and found no systematic relationship. These analyses argue against the possibility that a reduction in neural responsiveness after stroke could be an artifact of systematic changes in resting calcium levels. Collectively, these experiments and analysis indicate that chemogenetic therapy can prevent the stroke-related disruption of sensory responses in VIP neurons.

Since abnormal spontaneous cortical activity has been described in the first few hours to days after stroke[40–42], we imaged calcium transients in the absence of sensory stimulation for 75 s periods before stroke and at weekly intervals afterwards. Our experiments indicate that neurons would show periodic

calcium transients of variable amplitudes (Supp. Fig. 4a, b) that were generally similar or slightly larger in amplitude compared to sensory evoked responses. Overall, stroke in mice that did not receive chemogenetic treatment was associated with a trend towards fewer spontaneous calcium transients/events and significantly less cumulative event time in the first week after stroke (Supp. Fig. 4b, c), but not at later timepoints. In addition, we also considered the possibility that ongoing stroke induced spreading depolarizations (SD) could have influenced or contaminated our assessment of sensory responses at day 7. Therefore, we assessed SD events by recording slow direct current (DC) potentials or imaged large amplitude calcium events in the peri-infarct cortex within first 60–75 min after stroke or at 7 days recovery ($n = 3$ mice per group). Recording of DC potentials immediately after initiating photothrombosis revealed approximately 1–2 SD's occur within the first 30 min after stroke (Supp. Fig. 5a; 2 mice exhibited 2SD each whereas 1 mouse showed just 1SD; peak amplitude: $12 \pm 1.1$ mV; duration: $161.8 \pm 5.2$ s), which is consistent with previous studies[41]. However, we did not observe any SDs in peri-infarct cortex when recording at 7 days after stroke (Supp. Fig. 5a). To be extra careful, we also included a positive control experiment with 1 mM KCl application to prove that if a SD had occurred, we would have detected it. Similarly, by imaging SD induced calcium waves in peri-infarct cortex (which are ~10 times larger in amplitude than sensory-evoked calcium transients), we typically observed a SD wave within the first 30 min after stroke (Supp. Fig. 5b), but never 7 days post-stroke ($n = 3$ mice per group). Based on these experiments and previous literature, we conclude that it is very unlikely that ongoing SD waves could have affected our calcium imaging experiments and analysis collected 7 days after stroke.

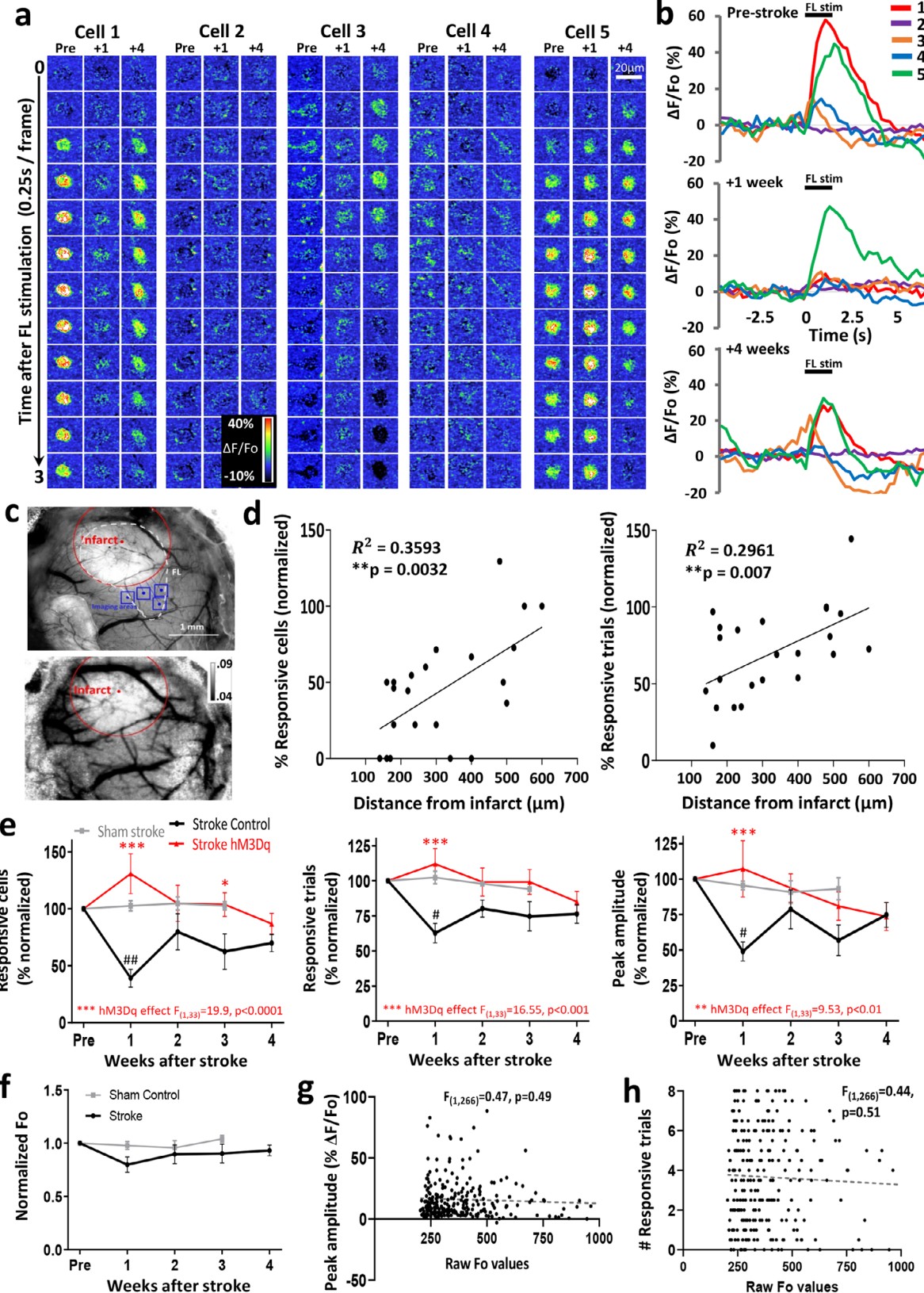

**Highly active VIP neurons are most susceptible to the effects of stroke and chemogenetic treatment.** As noted above, sensory responses were variable between neurons and trials. This raises the possibility that some VIP interneurons may be more active or hard wired into the forelimb sensory cortical circuit, while others may play little or no role. Furthermore, this variability raises the

interesting question of whether chemogenetic stimulation works by recruiting less responsive or inactive VIP neurons after stroke, or perhaps re-instates the function of those that were generally active and involved in forelimb sensory processing. To test these ideas, we parsed VIP neurons into one of three categories of "activity" based on the fidelity of their responses to forepaw

**Fig. 4 Disruption of sensory responses after stroke can be mitigated with chronic chemogenetic stimulation. a** Montages show forelimb-evoked calcium responses (1.5 s stimulation starting at 0 s) from a mouse that received control treatment in 5 different VIP interneurons before stroke (Pre), and 1, 4 weeks afterwards. Neurons were imaged ~23 h after last vehicle/CNO injection. **b** Forelimb-evoked calcium responses in five VIP neurons (average of eight trials from cells shown in Fig. 4a) before and after stroke. **c** Top: brightfield images show imaging locations (blue boxes) relative to the infarct. Bottom: laser speckle imaging reveals zones with poor blood flow (infarct) which exhibit higher speckle variance and thus appear lighter. Images are representative for all stroke mice in the study. **d** Scatterplots showing that the % responsive cells (left) and trials (right) for areas imaged 1 week after stroke (only in mice that received control treatment). Note that response disruptions are highly related to distance from the infarct border. **e** Effect of stroke and chemogenetic (hM3Dq) treatment on the % forelimb responsive cells, trials and peak response amplitudes in peri-infarct cortex (<400 μm of border) normalized to pre-stroke values. Data represent 116 neurons from 4 sham stroke mice, 154 neurons from 6 stroke mice given control treatment, or 197 neurons from 7 stroke mice that received chronic chemogenetic stimulation. Chemogenetic treatment prevented the stroke induced reduction in the fraction of responsive cells (two-way ANOVA, Main effect of hM3Dq: $F_{(1,33)} = 19.9$, $p < 0.0001$), responsive trials (two-way ANOVA, Main effect of hM3Dq: $F_{(1,33)} = 16.55$, $p = 0.0002$) and peak amplitude of responses (two-way ANOVA, Main effect of hM3Dq: $F_{(1,33)} = 9.53$, $p = 0.004$). **f** Normalized resting GCaMP6s fluorescence (Fo) in neurons show no significant changes over time (Sham Control = 4 mice; Stroke = 6 mice two-way ANOVA, Main effect of Group: $F_{(1,24)} = 3.9$, $p = 0.057$). **g** Two-sided linear regression analysis shows no relationship between resting calcium Fo values and the peak amplitude of sensory responses. **h** Two-sided linear regression analysis shows no relationship between number of responsive trials with resting calcium levels. ##$p < 0.01$, #$p < 0.05$ for $t$-test comparisons against pre-stroke. *$p < 0.05$, **$p < 0.01$, ***$p < 0.001$ for $t$-test comparisons between Stroke hM3Dq vs Stroke Control. Data show means ± S.E.M.

stimulation before stroke: high, moderate, and minimally active neurons (i.e. responds to forepaw stimulation in 6–8 trials, 3–5 trials, or 0–2 trials, respectively). We should note that highly and moderately active neurons also displayed significantly more spontaneous activity relative to neurons we defined as minimally active (Supp. Fig. 4d, e).

First we quantitatively assessed changes in the fraction of responsive trials (normalized to pre-stroke values) in high, moderate, and minimally active neurons. In the absence of chemogenetic treatment, stroke caused a major reduction in responsive trials in highly active neurons over time and to a lesser extent in moderately active neurons (Fig. 5a). These disruptions in responsiveness could be prevented by chemogenetic stimulation (see red asterisks in Fig. 5a). Next, we followed the trajectory of each category of neuron over time to determine if, for example, highly active neurons switched to a moderate/minimally active neuron or whether minimally active neurons could become highly active. In sham stroke mice, both highly and minimally active neurons tended to stay in their respective category over 3 weeks time whereas moderately active neurons could shift into high or minimally active categories (Fig. 5b). However for mice subjected to stroke that did not receive chemogenetic therapy, there was a significant drop in the % of highly active neurons (open bars) which shifted into the moderate (gray bars) or minimally (black bars) active category (Fig. 5b; Stroke Control vs Sham Stroke, $X^2 = 32.7$, $p < 0.01$). Interestingly, highly active neurons appeared most likely to recover in untreated stroke mice given that the percentage of highly active neurons increased from a low of 19% at 1 week, to 55% at 4 weeks (Fig. 5b). By contrast, highly active neurons in mice that received chronic chemogenetic stimulation were better able to remain highly active (Fig. 5b; Stroke hM3Dq vs Sham Stroke; $X^2 = 3.8$, $p > 0.05$), even at 1 week post-stroke when category switching was most likely to occur. However it should be noted that in rare cases, a neuron that was minimally active before stroke could become highly active to forepaw touch after stroke (Fig. 5c; 1/48 cells at 2 and 4 weeks, 3/48 cells at 3 weeks). And finally, we compared the percentage of highly, moderately and minimally active neurons within each experimental group before stroke versus afterwards. This analysis revealed no significant change in the proportion of each subpopulation in sham stroke or stroke affected mice that received chemogenetic stimulation (Fig. 5d). By contrast, stroke mice that received control stimulation exhibited significantly fewer highly active neurons and more minimally active neurons after stroke (Fig. 5d). In summary, these results indicate that stroke primarily disrupts the responses of highly active neurons

and increases the proportion of minimally active neurons, both of which can be mitigated with chronic chemogenetic stimulation. Further, our data argue against the idea that new forelimb responsive neurons are recruited after stroke.

One conceivable consequence of stroke is that it may wreak havoc on the predictability of neural responses to sensory stimulation, such that a population of neurons will respond with very different fidelities from week to week. Therefore we determined at baseline (pre-stroke/sham) whether a neuron responded to 0, 1, 2, 3, up to 8 trials. We then followed each neuron over time to determine what proportion of the population responded to 0, 1, 2...8 trials. From this, we could visualize the variability in the fidelity of responses of each population over time. As shown in Fig. 6a, several patterns can be visualized. First, populations in sham stroke mice showed less spread/variability in responses over time compared to stroke control mice (note the drift in colors over weeks in Stroke control vs Sham stroke in Fig. 6a). Second, response variability over time tended to be lower (ie. more predictable) for neurons at each end of the response spectrum (e.g. neurons at 0–1 or 8 responsive trials at BL). Third, stroke noticeably increased response variability in the highly active neurons in mice that received control treatment (made them less predictable) whereas those that received chemogenetic stimulation, were less disrupted and more predictable (Fig. 6a). In order to assess this phenomenon quantitatively, we calculated how much the population of cells at each time point deviated from pre-stroke values (e.g. if all cells at 1 week from the "8 Trial at BL" group responded to 8 trials, the deviation would be zero). This analysis indicated that in the absence of chemogenetic treatment, stroke led to a significant increase in response variability in highly active neurons, but did not affect moderate or minimally active neurons (Fig. 6b). Mice that received chemogenetic stimulation did not show the expected increase in variability within highly active neurons, as responses were similar to sham control mice (Fig. 6b).

## Discussion

Here we used longitudinal calcium imaging to examine how stroke affects sensory responses within VIP interneurons. Moreover we determined whether chemogenetic stimulation of these neurons could represent a strategy for promoting stroke recovery. By doing so, we have made the following findings. First we show that stimulation of VIP neurons with an excitatory DREADD can enhance weakened sensory responses in the stroke affected cortex and improve recovery of sensori-motor paw function. Second we find that the stroke disrupts sensory responses primarily within a subset of VIP neurons that were highly active/responsive before

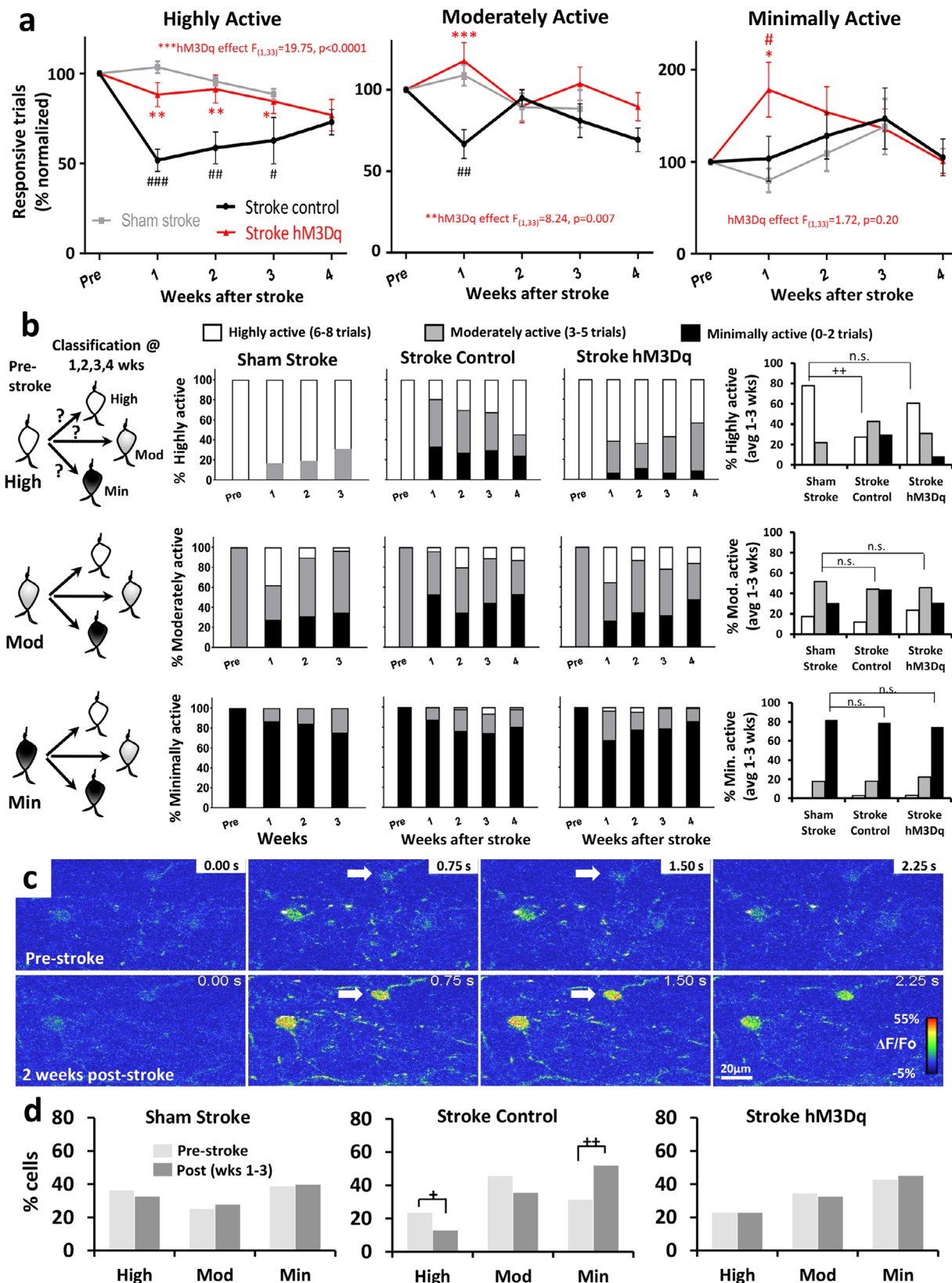

stroke. Of note, these neurons were also most likely to regain responses over time. Contrary to expectations based on previous stroke and barrel cortex plasticity studies, we did not find that stroke recovery involved the recruitment of minimally active VIP neurons. Importantly, preserving sensory response fidelity and predictability in these highly active VIP neurons with

chemogenetic therapy, was associated with improved stroke recovery (see summary in Fig. 7).

**Chemogenetic stimulation of disinhibitory VIP interneurons improves stroke recovery.** Work from our lab and others has

**Fig. 5 Highly active VIP neurons are most susceptible to the effects of stroke and chemogenetic therapy. a** VIP neurons were classified as highly, moderately or minimally active neurons based on the fidelity of their responses before stroke (highly, moderately and minimally active cell responds to 6–8, 3–5, and 0–2 trials respectively). Graphs show the effect of stroke ($n = 6$ and 7 mice for Stroke control and Stroke hM3Dq, respectively) or sham ($n = 4$ mice) procedure on the % forelimb responsive trials in each neuron subpopulation. Chemogenetic stimulation protects against the loss of response fidelity in highly active neurons (two-way ANOVA, Main effect of hM3Dq: $F_{(1,33)} = 19.75$, $p < 0.0001$), and to a lesser extent in moderately active neurons (two-way ANOVA, Main effect of hM3Dq: $F_{(1,33)} = 8.24$, $p = 0.007$), while there was no effect in minimally active neurons (two-way ANOVA, Main effect of hM3Dq: $F_{(1,33)} = 1.72$, $p = 0.20$). **b** To understand how response profiles can change over time, each neuron was assigned to one of the 3 categories and then followed over time after sham stroke (highly, moderately, minimally active = 42, 29, 45 neurons, respectively) or stroke with or without chemogenetic stimulation (Stroke hM3Dq = 44, 69, 84 neurons and Stroke control = 36, 70, 48 neurons). Far right: graphs show the proportion of each neuron class (out of 100%) averaged over weeks 1–3 relative to pre-stroke. **c** Montage showing forelimb-evoked calcium responses in a neuron (white arrow) that was minimally responsive before stroke, but became highly responsive afterwards. **d** Graphs comparing proportion (out of 100%) of high, moderate or minimally active neurons before stroke versus 1–3 weeks afterwards. Chi-squared ($\chi^2$) analysis revealed no significant changes in Sham stroke or Stroke hM3Dq groups (all $\chi^2$ values range from 0.01 to 0.38; all $p$ values > 0.05). Stroke control stimulation mice exhibited significantly fewer highly active neurons ($\chi^2 = 4.91$, $p < 0.05$) and more minimally active neurons after stroke ($\chi^2 = 13.73$, $p < 0.01$). $+p < 0.05$, $++p < 0.01$ for two-sided $\chi^2$ comparison. $\#p < 0.05$, $\#\#p < 0.01$, $\#\#\#p < 0.001$ for t-test comparisons against pre-stroke. $*p < 0.05$, $**p < 0.01$, $***p < 0.001$ for $t$-test comparisons between Stroke hM3Dq vs Stroke Control. n.s. not significant. Data show means ± S.E.M.

shown that cortical excitability in peri-infarct regions is disrupted by stroke. For example, fluorescence imaging or chronic multi-electrode recordings have shown that stroke dampens spiking activity, local field potentials and sensory/optogenetically evoked responses for several weeks after stroke[7,32,34,43–45]. There are likely many explanations for this deficit, such as lowered expression of GABA transporters[18] or de-afferentation of connections to and within the cortex[32,46,47]. Indeed, strategies aimed at overcoming these excitability deficits by chemically lowering extrasynaptic GABA inhibition[18,48], enhancing AMPA receptor mediated glutamatergic transmission[49] or optogenetic stimulation of excitatory connections in thalamus or cortex[33,50,51], have shown promise in promoting recovery in pre-clinical models of stroke. Although exciting, each approach has its own set of caveats that may limit applicability such as the need for invasive surgery or risks associated with brain wide manipulation of GABAergic or glutamatergic transmission (e.g. seizures, altered cognition). In the present study, we reasoned that chemogenetic manipulation of cortical excitability through VIP interneurons, could provide a focal boost in peri-infarct excitability. VIP neurons play a powerful role in enhancing sensory responses in pyramidal neurons in visual and auditory cortex through disinhibition[26–28,52,53]. Consistent with previous findings, we found that increasing VIP interneuron excitability with chemogenetic stimulation could enhance forepaw evoked responses in somatosensory cortex for at least 1 h after CNO injection. Based on this finding and previous data showing that the return of normal activity patterns in peri-infarct cortex supports behavioral recovery[45,54], we used chemogenetics to stimulate VIP neurons once a day from day 4 up to 6 weeks after stroke. We should note that chronic chemogenetic treatment was well tolerated by mice as they did not show any obvious changes in body appearance or display seizure activity. The benefits of this therapy on sensori-motor paw function became evident by 14 days recovery, which fits with previous studies showing that the sub-acute phase of stroke (~3–21 days post-stroke) is a critical time for enhancing recovery[55,56]. Importantly, these benefits persisted for weeks after treatment had stopped. Although we did not image the down-stream cellular targets of VIP neurons (e.g. Somatostatin, Parvalbumin, or pyramidal neurons), our VSD imaging experiments showed that responses in peri-infarct cortex were generally enhanced well after chemogenetic therapy had ceased. Since VSD imaging primarily reflects subthreshold depolarization in layer 2/3 excitatory neurons, this result implies that chemogenetic treatment induced a lasting and presumably adaptive change to cortical circuit excitability, beyond just VIP circuits. Precisely how these lasting downstream changes in circuitry are accomplished

remains an open question. Previous studies from our group and others have shown that stimulating cortical circuits with chemo- or opto-genetics can promote the proliferation and stabilization of axonal boutons, as well as enhance growth and plasticity associated signaling pathways (CREB, BDNF, NGF)[50,51,57]. Future slice electrophysiology studies that can address circuit specific changes in intrinsic excitability, spiking patterns, pre and post-synaptic forms of plasticity (LTP, LTD), would be informative to identify the synaptic mechanisms involved.

In addition to regulating cortical excitability through disinhibition, VIP interneurons have been implicated in regulating cerebral blood flow[58,59]. Given that stroke leads to long-lasting deficits in neurovascular coupling[34,60], it is conceivable that chemogenetic stimulation could have also influenced blood flow regulation. Although this is possible, our study did not reveal any detectable effect of chemogenetic stimulation on regional cerebral blood flow. We should note this does not necessarily exclude the possibility that sensory evoked changes in blood flow could have been preserved or enhanced with chemogenetic treatment. However given that longitudinal assessments of neurovascular coupling in the stroke affected brain are not trivial experiments, future studies will be needed to rigorously explore this idea.

**A subset of VIP neurons are highly sensitive to the effects of stroke and therapy.** To the best of our knowledge, our study is the first to longitudinally image and characterize sensory-driven responses in the same population of inhibitory interneurons before and several weeks after stroke. The power of this approach is that one can determine stroke and time dependent changes in sensory responses within the same neuron. Our imaging revealed that sensory responses in VIP neurons were heterogeneous. Accordingly, we classified VIP neurons into three groups based on the fidelity of their responses before stroke (high, moderate and minimally responsive) and followed each group over time. In sham stroke control mice, we found that about 45% of neurons were responsive to touch, and their response fidelity across trials was relatively stable. Some response variance was observed from week to week, although this is not unexpected based on previous imaging of excitatory neurons in somatosensory cortex[61]. However after the induction of stroke, we found that sensory responses in general were significantly dampened. This finding is consistent with previous acute and long-term imaging of putative pyramidal neuron populations after stroke[7,33,62]. Of note, this disruption in sensory response fidelity and predictability was more profound within a subset of highly active VIP neurons. Why this occurs is not known but could reflect the fact that they were most involved or hard wired in the sensory circuit before

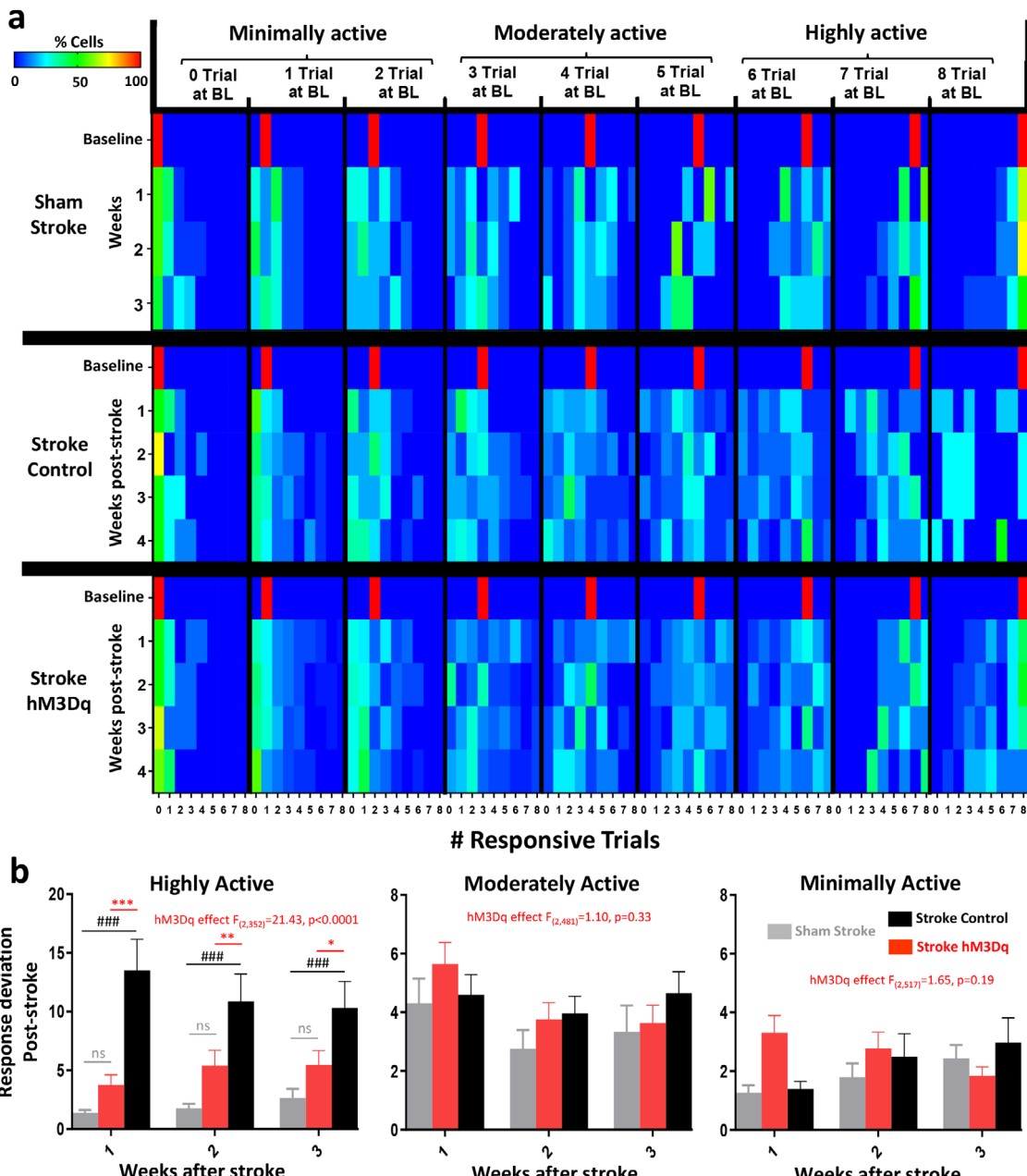

**Fig. 6 Chemogenetic stimulation preserves the predictable nature of sensory responses in highly active VIP neurons. a** Heat map illustrates changes in the % of cells (each population defined by the number of responsive trials at baseline) that responded to 0, 1, 2…8 stimulation trials each week. Note that variability in the population of responding cells is lowest for the least and most active population of neurons (e.g. see warm colors for population representing "0 or 8 trials at BL"). Stroke dramatically skews the response distribution in the highly active population which is mitigated with chemogenetic treatment. **b** Bar graphs quantify how much neurons in the highly, moderately, and minimally active population deviate from their baseline responses over time. Stroke mice that received control treatment showed significantly higher response deviation (ie. less predictability) in highly active neurons 1–3 weeks after stroke than mice that received chemogenetic stimulation or sham stroke controls (two-way ANOVA, Main effect of hM3Dq: $F_{(2,252)} = 32.71$, $p < 0.0001$; $n = 42, 44, 36$ neurons per group, respectively). There were no significant group differences in response deviation in moderate (two-way ANOVA, Main effect of hM3Dq: $F_{(2,481)} = 1.10$, $p = 0.33$; $n = 29, 69, 70$ neurons per group, respectively) or minimally active neurons (two-way ANOVA, Main effect of hM3Dq: $F_{(2,517)} = 1.65$, $p = 0.19$; $n = 45, 84, 48$ neurons per group, respectively). ###$p < 0.001$ for two-sided $t$-test comparisons against sham stroke. *$p < 0.05$, **$p < 0.01$, ***$p < 0.001$ for two-sided $t$-test comparisons between Stroke hM3Dq vs Stroke Control. n.s. not significant. Data show means ± S.E.M.

stroke, and therefore were most vulnerable to disruption. Future studies that image these cells in vivo and reconstruct their afferent inputs following stroke, might be revealing. What is rather surprising is the fact that minimally or moderately active cells did not, in sufficient numbers, become more active after stroke. Indeed, longitudinal imaging studies in barrel cortex suggested

that inactive neurons are recruited during spared whisker map plasticity[61]. Previous stroke studies from our lab and others have found that mesoscopic sensory and motor maps can shift or displace after stroke[36,63–65], which implies there might be remapping of function at a cellular level. Although Winship and Murphy convincingly showed that forelimb selective neurons

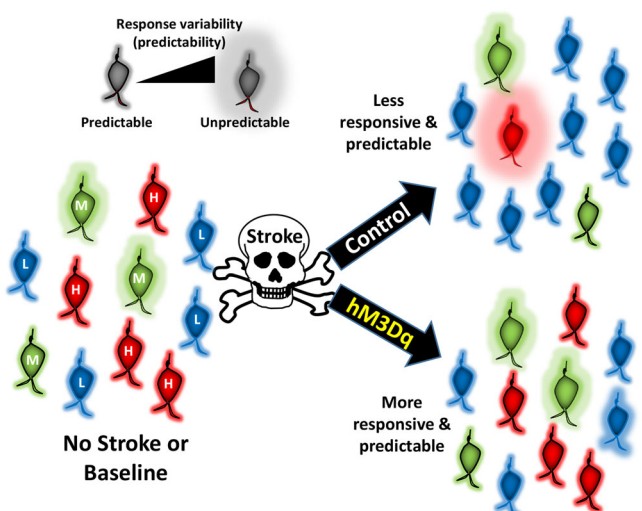

**Fig. 7 Summary diagram showing the effect of stroke and chemogenetic stimulation on VIP interneuron responses.** After stroke, there are fewer highly (H) responsive neurons and more minimal/low (L) responsive neurons in the peri-infarct cortex and their response became less predictable. Chemogenetic therapy preserves both the proportion and predictability of highly responsive neurons in peri-infarct cortex. M refers to moderately responsive neurons.

could become responsive to another limb after stroke, it was unknown that if same neurons could become more or less responsive to the same limb over time. Our work clearly shows that it is at least possible for minimally responsive neuron to become highly responsive after stroke, perhaps through some unmasking of inhibition or aberrant re-wiring after stroke. However, this occurred in only 1 or 2 neurons tracked over time and therefore was quite rare. What we do generally find is that highly active neurons that lose responsiveness after stroke, are the same cells that regain responsiveness over the 4 week recovery period. This finding fits very well with our previous thalamo-cortical calcium imaging study[33] and a new elegant imaging study from the Portera-Calliau lab showing that stroke-related plasticity of excitatory layer 2/3 neurons involves re-activating existing circuits rather than assigning inactive circuits new roles[62]. When taken together, these findings indicate that stroke recovery and map plasticity after stroke are likely the product of both sub-cortical and cortical excitatory and dis-inhibitory circuits regaining their normal activity/response patterns. Why stroke-related plasticity usually involves the same circuits is unknown, but it could reflect the transient loss of pre-synaptic inputs that must re-innervate familiar post-synaptic targets. Indeed, recent work has shown that when dendritic spines retract after ischemia, they remain attached to axonal boutons[66] which could provide a structural explanation for the temporary disruption and re-activation of highly active circuits. Alternatively, it could reflect temporary edema and inflammation that resolves 1–2 weeks after stroke. One obvious implication of these findings is that it is absolutely critical to preserve at least some of these highly active circuits, which can then be targeted by therapies for regaining function.

By following functional subpopulations of VIP neurons over time, we also discovered that chemogenetic therapy could restore sensory responses in highly active neurons, even by 1 week recovery. This means that 3 consecutive days of chemogenetic treatment before imaging at day 7 was sufficient to restore these responses. The stimulation effect of CNO injection only lasts a

few hours each day and therefore it is very unlikely there was some residual effect 24 h later when imaging commenced. It is possible that chemogenetic stimulation prevented the delayed functional weakening of these circuits from day 4 to 7, although this is less likely since a recent study demonstrated that sensory responses are already weakened by day 5 after stroke[62]. Rather, we suspect that chemogenetic stimulation very rapidly promotes the functional re-activation of these putatively weakened circuits. Precisely how this is accomplished, perhaps through enhancing Hebbian/spike timing dependent synaptic plasticity[9], will require further study. What is interesting is the fact that enhancing the function of a subpopulation of interneurons, preceded improvements in forepaw sensori-motor abilities. Indeed, the enhanced recovery of forepaw abilities in stimulated mice was not evident until the 2nd week after stroke whereas VIP circuits were functionally back "on-line" at 1 week. Perhaps this sequence allows for the progressive re-activation of excitability in targets downstream of VIP neurons such as excitatory pyramidal neurons, which ultimately, may best correlate with the return of sensori-motor abilities.

The idea that activating a subset of interneurons (such as those that express VIP or subpopulations therein), could play an influential role in stroke-related cortical plasticity and recovery, is not without precedence. Previous studies that have selectively activated or ablated somatosensory cortical neurons have shown that remarkably few neurons (~10–300 neurons) are needed for sensory perception and learning[67,68]. VIP neurons are especially abundant in cortical layer 2/3[35,69] and receive sensory information from the thalamus[25]. These neurons are also highly integrated with motor circuits, where they can reciprocally regulate activity levels[52,70]. Our study and previous ones show that upregulating VIP interneuron excitability can enhance sensory-evoked layer 2/3 population responses in somatosensory, visual and auditory cortex[26–28], as well as the tuning of specific sensory features[71]. Conversely, interfering with VIP neuron activity impairs visual response selectivity, learning, and ocular dominance plasticity[29,30]. In the context of stroke, chronic upregulation of VIP interneuron excitability with chemogenetics may increase the gain of weak sensory or motor forelimb signals, similar to that described in visual cortex[27,52]. Amplification of these weak signals may be important for functional recovery, especially during the first few weeks after stroke when thalamic and intracortical excitability are maximally impaired. It is also interesting to note that chemogenetic therapy was associated with greater predictability (lower week to week variability in responses) of sensory responses in highly active neurons. While we do not know the functional significance of this effect, it is conceivable that unreliable or unpredictable activation of somatosensory neurons may ultimately degrade tactile perceptions, therefore preserving them is key for recovery.

**Potential limitations.** Our study, while comprehensive in its breadth of approaches for studying the influence of stroke on VIP interneuron function and how they can affect recovery, has its limitations. First, we do not yet know if highly active VIP interneurons have a specific identity since it is increasingly recognized that VIP interneurons are diverse in their molecular, functional, and anatomical characteristics[17]. In our study we classified neurons based on the fidelity of their responses to sensory stimulation before stroke. It is interesting to note that the effects of stroke (without chemogenetic treatment) on each subgroups response fidelity and predictability were noticeably different (see Figs. 5 and 6), perhaps strengthening the argument that there are inherent functional differences between subgroups. Further, our

analysis of spontaneous activity clearly showed a separation between high/moderately active versus minimally active neurons (Supp. Fig. 4), with the latter showing low levels of activity at rest. One interesting possibility is that high or moderately active neurons may represent burst spiking VIP neurons that are susceptible to neuromodulatory factors and have been specifically identified in somatosensory layer 2/3[72]. Another possibility is that the highly active VIP neurons could be the same ones that possess a high input resistance, preferentially contact interneurons and express calretinin[73]. However, post-mortem identification of highly active VIP neurons (imaged in vivo) that co-localize with calretinin, did not reveal a clear distinction in the response profile of VIP neurons that express calretinin versus those that do not (Supp. Fig. 6a–c). Future studies that combine calcium imaging with electrophysiology, single cell transcriptomics and/or post-mortem staining of proteins that associate with VIP neurons (calretinin, CCK, ChaT), would be needed unravel this mystery. Another potential limitation is our use of light anesthesia during calcium imaging. Based on our previous experience imaging awake and anesthetized mice, we reasoned that reliable and replicable delivery of our vibro-tactile stimulus was our top priority, especially if we wanted to compare forelimb sensory responses across mice and treatment groups. Unfortunately awake mice constantly make volitional or respiratory tapping movements involving the forepaw, which greatly complicates the interpretation of pure sensory responses to a stimulus, and requires us to discard many stimulus trials. While not a perfect comparison, our pilot experiments imaging VIP responses to vibro-tactile stimulation of the forepaw in awake mice, suggest a slightly elevated but otherwise comparable response profile (Supp. Fig. 3). Given these findings, we can be reasonably confident that anesthesia did not completely perturb VIP neuron sensory responses.

In conclusion, our study enriches current thinking on how stroke and a chemogenetic therapy can influence the function of disinhibitory VIP interneurons. Our data indicate that the disruptive effects of stroke are not evenly distributed, as those that are more active/responsive to sensory stimuli before stroke, are particularly vulnerable afterwards. Moreover our data show that the recovery of sensory responses, be it slowly as seen with spontaneous recovery in untreated mice, or quickly when induced with chemogenetic stimulation, occurs primarily in this sub-population of highly active neurons and does not involve de novo recruitment of neurons not previously engaged in sensory responses. These findings will inform future studies that seek to refine and optimize treatment strategies for improving functional recovery, perhaps by targeting specific subpopulations of neurons.

## Methods

**Animals**. In all, 2–5-month-old male and female VIP-IRES-cre mice (Jackson Laboratory, stock no. 010908) were used in this study. Male mice were used for electrophysiology, VSD and calcium imaging experiments, as well as the first cohort of behavioral studies testing the chemogenetic therapy (see Fig. 2a; cohort 1, K.G. 2014), whereas female mice were used for the second cohort of behavioral studies (cohort 2, S.C. 2018). As a general rule, littermates were randomly assigned to experimental or control conditions by flip of the coin. Mice were housed in standard laboratory cages in groups of 2–5 under a 12-h light/dark cycle and given access to ad libitum water and food. Room temperature was kept between 21 and 24 °C and between 40 and 60% relative humidity. All experiments were conducted in accordance with the guidelines laid out by the Canadian Council of Animal Care and approved by the University of Victoria Animal Care Committee.

**Cranial window and AAV injection**. Surgical implantation of glass cranial windows was performed as previously published[33,74]. Briefly, mice were anesthetized with isoflurane (2% induction, 1.5% maintenance) mixed in medical air and fitted into a custom-made surgical stage. During the procedure, animals were kept on a heating pad and their body temperature was maintained at 37 °C. Animals received 0.02 mL injection of dexamethasone (0.2 mg/kg, s.c.) to reduce any surgery-induced inflammation during and after the procedure. A metal ring designed to

hold the head during imaging (outer diameter 11.3 mm, inner 7.0 mm, height 1.5 mm) was secured to the skull (cyanoacrylate glue) centered over the FL somatosensory cortex. Using a high speed drill, one or two small holes were drilled through the skull for AAV injections. For each injection, 0.2–0.4 μL volume of AAV (purchased from Addgene) mixed with HEPES-buffered ACSF was pressure injected into the cortex using a glass micropipette with the tip diameter of ~30-50 μm connected to a Hamilton syringe. AAVs injected into the cortex alone or in combination were: a) AAV2.hSyn.DiO.hM3d(Gq).mcherry (1:10 dilution; gift from Bryan Roth, Addgene viral prep #44361-AAV2), b) AAV1.CAG.-Flex.eGFP.WPRE.bGH (1:50 dilution), c) AAV1.Syn.Flex.GCaMP6s.WPRE.SV40 (1:10 dilution; gift from Douglas Kim and GENIE Project, Addgene viral prep #100845-AAV1). Following AAV injection, a 4 mm diameter craniotomy was drilled in the center of metal ring. Cold HEPES-buffered artificial cerebrospinal fluid (ACSF) was intermittently applied to the skull during the drilling procedure to keep the brain moist and cool. The thinned portion of skull bone was removed (leaving the dura intact) and was covered with a circular glass coverslip (no.1 thickness). The coverslip was affixed to the skull using cyanoacrylate glue and dental cement. Following the procedure, mice were allowed to recover under a heat lamp and transferred to their home cage. After four weeks recovery from the surgery, the quality of imaging windows was examined and mice whose windows with a significant loss of clarity were excluded from the study.

**Intrinsic optical signal imaging**. IOS imaging was performed 4 weeks after cranial window implantation to map cortical areas corresponding to FL and HL somatosensory cortex. The cortical surface of lightly anesthetized mice (1% isoflurane) was illuminated with a red LED (635 nm). Sensory-evoked changes in the reflectance of red light, attributable to changes in the levels of deoxy-hemoglobin and presumably neuronal activity, were collected with a MiCAM02 CCD camera (SciMedia) mounted on an upright microscope through a ×2 objective. The plane of focus was set ~200 μm below the brain's surface to minimize the contribution of hemodynamic signals from large surface vessels. In order to evoke cortical responses, 1 s of vibro-tactile stimulation (5 ms bi-phasic pulses vibrating at 100 Hz) was applied to the contralateral FL or HL through a pencil lead connected to a piezo-electric element. Each imaging session consisted of two sets of 12 stimulation/no stimulation trials with a 10 s interval between each trial. In each trial, intrinsic signals from the cortical surface were recorded over 3 s period (1 s of pre- and 2 s post-stimulation) at 100 Hz frame rate with 10 ms exposure time. Twelve trials were then mean filtered (5 pixel radius) and averaged together to generate an average stack. Relative changes in reflectance of red light resulting from the stimulation ($\Delta R/Ro$) were then calculated by normalizing all images to an average intensity projection of pre-stimulus images. Subsequently, an average intensity projection of images during the post-stimulus response (0.5–2 s) was taken. This average response image was then thresholded (at 75% of maximum intensity) and used to superimpose the IOS map of FL and HL somatosensory areas on an image of the cortical surface vasculature.

**Photothrombotic stroke in forelimb somatosensory cortex**. A photothrombotic stroke was targeted to the forelimb somatosensory cortex in the right hemisphere through the cranial window[75]. Briefly, mice were anesthetized using isoflurane (2% induction; 1.5% maintenance) mixed with medical air and placed under an Olympus BX51WI microscope. Body temperature was maintained at 37 °C. Animals received an i.p injection of 1% Rose Bengal solution (100 mg/kg in HEPES-buffered ACSF). A 1 mm diameter area of the FLS1 cortex (using maps acquired from IOS imaging) was illuminated with a green LED (~20 mW) through a ×10 objective lens for 15–20 min until surface vessels had clearly stopped flowing. The photo-activated region was positioned in close proximity (~200 μm) to previously defined pre-stroke imaging areas. Sham stroke mice received either Rose Bengal injection (without illumination of green light) or illumination of green light (without the injection of photosensitive dye).

**Electrophysiological recording of sensory-evoked responses or spreading depolarization**. Mice were lightly anesthetized with 15% urethane dissolved in water (1.25 g/kg). Once tail pinch reflexes were lost, mice snugly secured into a surgical plate with body temperature clamped at 37 °C. A small hole was drilled through the skull above the right FLS1 cortex so a 1–2 MΩ glass micropipette filled with HEPES-buffered ACSF could be inserted into the brain 200–300 μm below the cortical surface. Evoked potentials were amplified (1000x) and filtered between 1-1000 Hz with a differential amplifier (A-M Systems). A single 5 ms deflection of the forepaw with a piezoelectric wafer was used to evoke cortical field potentials every 10 s and averaged over 45 trials. Cortical responses were collected for up to 90 min after injection of vehicle and/or CNO (0.3–0.5 mg/kg, i.p.). The effects of 0.3 mg/kg CNO ($n = 5$ mice) on cortical responses in DREADD expressing mice was initially collected by K.G. and later replicated by M.M. using a dose of 0.5 mg/kg CNO ($n = 4$ mice). Since the percentage change in response amplitudes were similar for both doses of CNO, data were binned together. For analyzing the effects of CNO or vehicle on response amplitudes, the peak amplitude from a trial bin (average of 45 stimulation sweeps over 7.5 min) collected 30 min after injection was normalized to the peak amplitude at baseline.

Recording of slow direct current (DC) potentials to detect spreading depolarization in peri-infarct cortex were collected for 75 min immediately after stroke or at 7 days recovery. Glass recording pipettes filled with saline (~1–2 MΩ resistance) were inserted into peri-infarct cortex in urethane-anesthetized mice. Cortical potentials were amplified (10X), filtered at 0.1 kHz using a Getting Microelectrode Amplifier (Model 5, Iowa City, IA), digitized at 1 kHz with Digidata 123 A interface board and analyzed with pClamp10 software (Clampex, Version 10.2.0.12).

**Assessment of cerebral blood flow**. As previously described[76], laser doppler flowmetry was used to assess the effects of CNO on regional cerebral blood flow. Urethane-anesthetized mice had a 3 mm diameter optical probe lowered to ~1 mm above the thinned skull over the right somatosensory cortex (Moor Instruments, MoorVMS-LDF1, PC version 2.2). Perfusion measurements were sampled at 2 Hz over a 60-90 min period. Following 10-15 min of stable baseline measurements, mice were injected with vehicle and/or CNO (0.3–0.5 mg/kg, i.p.). Perfusion measurements collected after injection were normalized to the average baseline value. To determine the effect of saline/CNO injection or $CO_2$ inhalation on cerebral blood flow, perfusion units were averaged over 5 min period starting 15–20 min after injection, or 3 min after initiating CO2 inhalation and expressed relative to baseline.

**Voltage sensitive dye imaging**. Ten weeks after the induction of stroke, mice were anesthetized with 1% isoflurane, fitted into a custom built stereotaxic frame where body temperature was maintained at 37 °C. To prevent any movement during imaging, the skull was secured to a metal plate using cyanoacrylate glue and dental cement, which was fastened to the surgery stage. Mice were administered 0.12 mL of 20 mM glucose dissolved in water every 2 h to maintain proper hydration and glucose levels. A 5 mm diameter portion of the skull overlying the right FLS1 cortex was thinned, the skull carefully lifted off and the dura removed. The cortex was then bathed in RH1692 dye[77] dissolved in HEPES-buffered ACSF for 75–90 min (1 mg/ml passed through 0.22 μm syringe filter). After staining, the cortical surface was washed with ACSF several times to remove unbound dye. The brain was covered with 1.3% low-melt agarose dissolved in a HEPES-buffered ACSF and sealed with a glass cover slip.

For imaging, RH-1692 dye was excited with a high powered red LED (627 nm, ~20 mW at back aperture) that was passed through a Cy5 filter cube (exciter: 605–650 nm, emitter: 670–720 nm). Red light was focused on the cortical surface and collected using an Olympus XFluor 2X objective (NA = 0.14). A high speed SciMedia MiCAM02 CCD camera running Brain Vision software (Brain Vision BV-Ana X86 edition, Version 12.08.20) collected 12-bit image frames (184 × 124 pixels) every 4 ms. The left (impaired) forepaw or hindpaw was mechanically deflected by a single 5 ms pulse from a pencil lead connected to a piezoelectric wafer (Q220-AY-203YB, Piezo Systems; ~300 μm deflection). Auditory evoked cortical responses were eliminated by occluding both ears with low-melt agarose and Vasoline. For each trial, images were collected 250 ms before a single deflection of the forepaw and then 550 ms afterwards. To correct for dye bleaching, stimulation trials were divided by null stimulation trials. This process was repeated 12 times for each condition with a 10 s interval between each stimulation. Cortical depolarizations are expressed as the percent change in VSD fluorescence (ΔF/Fo) relative to pre-stimulation fluorescence (100 ms before stimulation). Montages of cortical responses were generated by mean filtering ΔF/Fo image stacks (radius = 2) and then binning two 4 ms frames in time. Forelimb-evoked depolarizations were analyzed from ΔF/Fo image stacks using ImageJ software (FIJIv1.52p; Java1.8.0_172-64bit). Responses in peri-infarct cortex were quantified in six regions of interest (each ROI = 0.16mm$^2$) that encircled the infarct, except for the anterior-lateral quadrant which was always close to the edge of the cranial window and difficult to reliably measure signals from. The peak amplitude, time to peak amplitude and half-width (ie. duration) of forelimb-evoked signals in the first 150 ms after stimulation were measured with Clampfit 9.0 software (Molecular Devices).

**Laser speckle imaging**. One week after stroke, the cortical surface was illuminated with a 785 nm elliptical laser beam 2.4 × 3.4 mm coupled to a 3X beam expander (ThorLabs Inc; 1–3 mW output power). Twelve-bit images were collected with a CCD camera mounted on an Olympus microscope (running Q-capture Pro 7 software) through a ×2 objective (696 × 520 pixels). Each imaging trial consisted of 100 consecutive frames of laser speckle images (exposure time T = 10 ms). Using ImageJ software (FIJIv1.52p; Java1.8.0_172-64bit), we first generated an average projection of all original images in each trial. The original image stack was processed with a two-dimensional variance filter (radius = 1 pixel) followed by taking the square root (to calculate standard deviation in images), then average projecting all image frames. This average image projection was then divided by the original average image projection (AVG SD/AVG Mean) to create a speckle contrast image. This image provides a visual representation of relative blood perfusion over the brain surface where speckle contrast is inversely related to blood flow (lower pixel values corresponding to higher blood flow and higher pixel values corresponding to lower blood flow). Speckle contrast images were thresholded to 80% of maximum intensity in the infarct region. The infarct region was subsequently enclosed by a circular ROI and mapped onto an image of the cortical surface. The distance of

each area relative to the infarct border was calculated as a straight line segment from the center of area to the infarct's circular border.

**Behavioral testing**. Sensori-motor function of the forepaw was assessed using the horizontal ladder walking and tape removal tests. Tests were administered over three sessions prior to stroke to establish a baseline level of performance, then 1 day after stroke followed by once weekly thereafter. During the treatment period, mice were always tested before (usually 1–2 h) receiving vehicle or CNO injection, to avoid being under the acute influence of the treatment. For each session of testing, mice underwent three trials per task. For the ladder walking test, mice were videotaped from below as they crossing the horizontal ladder with unevenly spaced rungs (70 cm long, 1–2 cm spaces between rungs, 1 mm diameter rungs). Steps were visualized using slow-motion video playback and scored by an observer blind to condition as one of three categories: slip, partial, or correct. A correct step occurred when the mouse placed its paw centered on the rung, so that the weight was supported by the palmer surface of the paw. A partial step was one where the paw stayed on the rung, but was placed so that either the heel or the toes was touching the rung. A slip was categorized as a step that either completely missed or a step that was touched the rung but then slid off, causing a slight fall. The fraction of each type of step was estimated by dividing by the total number of steps.

For the tape removal test, each trial was initiated by placing 5 mm diameter circles of adhesive medical tape onto the palmar surface of both forepaws and placing the mouse into a clear glass cylinder. The mouse was filmed from below and allowed up to 2 min to remove the pieces of tape. A blind observer recorded latencies to remove each piece of tape using slow-motion video playback.

**Histology and confocal imaging**. For experiments where VIP neurons imaged in vivo were re-identified in post-mortem brain sections, lysine fixable Texas Red dextran (Invitrogen, D1864) was injected intravenously and single capillaries were ruptured at the edge of each imaging area to create fiducial landmarks. Mice were then overdosed with sodium pentobarbital and perfused intracardially with 0.1 M PBS followed by 4% paraformaldehyde in PBS. Brains were post-fixed at 4 °C overnight and sectioned into 50 μm thick coronal or tangential sections using a freezing microtome. For quantifying infarct volume, every third cresyl violet stained section was imaged under brightfield with a ×4 objective (NA = 0.13) and an experimenter blind to condition traced the infarct zone using ImageJ software. The infarct volume was calculated by summing up the infarct area for each section multiplied by the distance between each section.

Immunostaining for mcherry tagged hM3Dq expressing VIP neurons or Calretinin was achieved by incubating free floating fixed sections in Rabbit anti-mCherry (1:1000 dilution, Novus Biologicals) or Rabbit anti-calretinin (1:2000, Sigma AB5054) in 0.1 M PBS with 0.1% TX-100 overnight at room temperature. Sections were washed, incubated in Cy5 conjugated Goat anti-Rabbit cross-adsorbed secondary antibody (Invitrogen A10523; dilution 1:500) for 4 h, mounted and coverslipped with Fluormount-G (Thermo-Fisher). An Olympus confocal microscope equipped with a ×10 objective lens (NA = 0.4) was used to acquire images of GCaMP6s, GFP, calretinin or hM3DGq mcherry expressing VIP neurons. Image stacks collected from somatosensory cortex were sampled at 4 μm z-steps at a pixel resolution of 1.24 μm/pixel.

**In vivo calcium imaging**. Two-photon imaging was used to examine the response properties of GCaMP6s-expressing VIP cells. Mice were lightly anesthetized with isoflurane (0.9–1% mixed in medical air) and fitted into a custom-made stage that stabilized the head under the objective. During imaging, mice were kept on a heating pad and their body temperature was maintained at 37 °C. Two photon images were acquired through the cranial window using an Olympus FV1000MPE galvanometric laser scanning microscope and Olympus software (FV1000-Version03.01) equipped with a mode locked Ti:sapphire laser. Images were collected through a 20x Olympus XLUPlanFl water-immersion objective lens (NA = 0.95) covering a 317.331 × 317.331 μm field of view (0.619 μm/pixel). GCaMP6s was excited with the laser wavelength tuned to 940 nm and power ranged from 20 to 60 mW at the back aperture depending on imaging depth.

Two-photon calcium imaging was conducted at weekly intervals before and after stroke, except for our experiments quantifying ischemia induced spreading depolarizations (imaging done within first 60 min and 7 days after stroke). Due to the fact that ischemic stroke damage could not be controlled with micrometer precision, some neurons imaged before stroke were destroyed. Thus, imaging after stroke was focused on "peri-infarct" regions that contained viable neurons which would persist for the remaining 4 weeks of study. The viability of these peri-infarct neurons was expected given previous studies showing relatively normal blood flow in peri-infarct regions by 5–7 days recovery[78,79]. Sensory-evoked calcium transients in VIP neurons were measured in response to 1.5 s of 100 Hz vibro-tactile stimulation of the left forepaw. Simulation was delivered to the dorsal surface of the forepaw through a pencil lead connected to a piezoelectric bending actuator (Piezo Systems, Q220-A4-203YB; 300 μm deflection). Sensory-evoked calcium responses were recorded for 8 stimulation trials for each imaging area (trials were recorded iteratively with one minute interval between them). Images for each trial were collected at 4 Hz with 5 s of pre-stimulus and 7 s of post-stimulus data acquisition. Spontaneous calcium transients were also recorded over a 75 s period of time in the

absence of any sensory stimulation. Imaging for spreading depolarization was collected at 1 Hz for 30–60 min immediately after photothrombotic stroke or for a period of 30-60 min at 7 days recovery.

**Analysis of calcium imaging data.** Individual trials were corrected for misalignments in x-y plane (possible drifts in brain position that occur during in-vivo imaging) using automated plug-ins StackReg (http://bigwww.epfl.ch/thevenaz/stackreg) and TurboReg (http://bigwww.epfl.ch/thevenaz/turboreg) in ImageJ software[80]. All images were registered to the first imaging trial. In order to identify GCaMP6s-expressing cell bodies for analysis, 8 stimulus trials were averaged together for each imaging area. Regions of interests (ROIs) were manually drawn around each cell body in each imaging trial and raw calcium signals were extracted by averaging all the pixels within each ROI. Raw calcium transients were adjusted for potential signal contamination emanating from labeled neuropil. The neuropil signal Fneuropil $(t)$ surrounding each cell was measured by averaging the signal of all pixels within a circular doughnut-shaped band, 10 μm wide, around the cell body excluding its processes and neighboring cells. The fluorescence signal of a cell body was estimated as Fcell-true $(t) =$ Fcell-measured $(t) - r \times$ Fneuropil $(t)$ where $t$ is time and $r$ is the correction ratio estimated as 0.7 for 0.95 numerical aperture[81–84]. Neuropil corrected forelimb-evoked calcium transients in each trial (Fcell-true or "F") were subtracted and normalized to their pre-stimulus signal ("Fo") to generate a ΔF/Fo {(F−Fo)/Fo} where Fo was the median value of fluorescent signal over 5 s before stimulation. Calcium transients for each trial were identified to be forepaw responsive if they demonstrated both: (a) significant stimulus related changes in ΔF/Fo based on a two-tailed Student $t$-test assuming unequal variances comparing 2.5 s following stimulus to 2.5 s preceding stimulus and (b) peak ΔF/Fo values within 2.5 s after stimulation were >10% pre-stimulus values. Forepaw responsive cells were subsequently defined as cells with more than one responsive trial. Given the longitudinal nature of this imaging study, cells that were not identifiable over weeks were excluded from the analysis. Additionally, cell bodies that were not at least 5% brighter than their neuropil (based on median value of soma vs surround neuropil) were not included in the analysis. Neuronal responsiveness was evaluated by comparing the fraction of responsive cells, fraction of responsive trials and average response amplitude (ΔF/Fo).

Similar to the analysis of sensory-evoked responses, spontaneous calcium traces were extracted from each cell body and corrected for potential neuropil contamination. ΔF/Fo was calculated for each calcium trace where Fo was set to be the 30th percentile value of each individual trace. The standard deviation of all data points smaller than the 30th percentile value of each trace was calculated (SDnot-event-related). In each calcium trace, data points that exceeded 10% of Fo value and were significantly different from not-event-related data points (> Fo+2SDnot-event-related) were considered a spontaneous calcium event. Only data points that persisted to meet the aforementioned criteria for at least six imaging frames (1.5 s), given the slow decay of GCaMP6s signals, were considered to be event related. Otherwise, they were assumed to be noise-like signal fluctuations. Subsequently, the total number of individual events and the total time associated with all calcium events in each recording were calculated and used as a measure of spontaneous activity.

In order to examine the variability of responses across weekly imaging sessions, deviations in response fidelity were measured for individual neurons. Deviations in response fidelity were quantified by calculating the squared differences of the number of responsive trials for each cell during each week's imaging session from those values collected during the Pre-stroke (stroke control, stroke hM3Dq) or Baseline (control) imaging session. These squared differences were then compared between stroke control, stroke hM3Dq, and sham control groups.

**Statistics.** Statistical analysis of the data was conducted using Microsoft Excel or GraphPad Prism 8 software (version 8.4.3). Data presented in graphs are means ± standard error of the mean (S.E.M.). Sample sizes were based on comparable n-values from previously published literature[7,32,33,43,44]. Most analysis involved a two-way analysis of variances (ANOVA) to identify significant differences between groups, time effects, and group by time interactions. Multiple comparisons were followed up with Fisher's LSD test. For between group comparisons involving a single factor, a one-way ANOVA (e.g. effect of CNO on Blood flow) or unpaired t-test was used (e.g. infarct volume). Chi-squared analysis was used to analyze changes in the proportion of VIP neurons with high, moderate or minimally activity. The "expected" values for Chi-squared analysis were derived from Sham stroke (Fig. 5b) or Pre-stroke values (Fig. 5d). The "observed" values were derived from an average of values from post-stroke weeks 1–3. The chi-squared statistic was calculated by squaring the difference between observed and expected values, divided by the expected value. $P$-values < 0.05 were considered significant for all tests. * indicates significant comparisons (*$p$ < 0.05; **$p$ < 0.01; ***$p$ < 0.001).

**Reporting summary.** Further information on research design is available in the Nature Research Reporting Summary linked to this article.

## Data availability

Data generated for this study are available from the corresponding author on reasonable request. Source data are provided with this paper.

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

## Acknowledgements
We are grateful to Dr. Pat Reeson for his advice and Taimei Yang for managing the mouse colony. Work was supported by operating, salary and equipment grants to C.E.B. from the Canadian Institutes of Health Research (CIHR), Heart and Stroke Foundation (HSF), and Natural Sciences and Engineering Research Council (NSERC).

## Author contributions
C.E.B. conceived of the study and supervised the trainees. M.M., K.G., and C.E.B. co-wrote the manuscript. M.M., K.G., S-E.C., and C.E.B. performed experiments, collected, and performed data analysis. R.B. and E.W. performed mouse surgeries. K.R.D. provided technical assistance and advice for electrophysiology and calcium imaging experiments.

## Competing interests
The authors declare no competing interests.
