## [Peer Review File · Nature Communications]

Reviewers' Comments:

Reviewer #1:

Remarks to the Author:

Motaharinia et al. explored whether chemogenetic modulation of VIP+ interneurons in the somatosensory cortex of mice can be beneficial for stroke recovery. The authors provide very interesting data with longitudinal imaging of VIP+ interneurons before and several weeks after local stroke induction. The major finding is that driving VIP+ cells chemogenetically can rescue cortical excitability after stroke and fasten functional recovery. The data is of highest quality and were obtained in carefully designed experiments. I only have several minor comments/questions to clarify some findings.

1. The manuscript gives an impression that the major role of neocortical VIP+ cells is disinhibitory; however, this may not be the case. VIP+ neuronal population is highly heterogeneous, with up to 80% of VIP+ contacts formed onto GABA⁻ structures (Zhou et al., 2017). This heterogeneity may result in highly variable patterns of activity observed by the authors in VIP+ cells, and needs to be carefully discussed. For example, across cortical areas, VIP+/CCK+ basket cells contact PYRs, whereas VIP+/CR+ cells contact preferentially interneurons (Guet-McCreight et al., 2020). Importantly, these two VIP+ subtypes have different intrinsic excitability, with VIP+/CR+ cells showing much higher input resistance. Accordingly, the disinhibitory VIP+/CR+ cells may represent the highly active group in this study and a preferential target for post-stroke therapy. It seems the authors are able to find VIP+ cells imaged *in vivo* for post-mortem IHC identification (Fig. 3B). It would be therefore possible to check whether the highly active group is expressing CR.

2. Was there any difference in evoked field response for 0.3 vs. 0.5 mg/kg of CNO? Why not using the same dose in all experiments? Ideally, one would use the lowest efficient dose throughout the entire study.

3. Why the therapy had started 4 days after stroke? Would a faster intervention be more beneficial? Have the authors tried different time intervals after the stroke, like 1, 2, 3 days?

4. Were there any acute effects of CNO treatment?

5. Was the beneficial effect of chemogenetic treatment persistent? Or have the authors evaluated the state of animals at a longer scale – 3-4 weeks (up to a month) after therapy cessation?

6. It is exciting to see that a single dose of CNO, with VIP activation time course of about 2 h (max up to 10 h), produces such a long-lasting (at least 23 h) effect on cortical excitability. Can you discuss a bit more the possible mechanisms of this plasticity?

7. In Discussion, the authors propose that chemogenetic manipulation can represent a relatively non-invasive way for boosting excitability. I would disagree with this point, as it requires stereotaxic surgery to deliver the viral vectors expressing the DREADD. The obvious advantage of this method is, however, its “targetability” or a possibility of focal manipulations with a local neuronal circuits in peri-infarct area.

8. Given that male and female mice were used in the study, were there any sex-specific differences in VIP+ cell activity or other reported phenomena?

Reviewer #2:

Remarks to the Author:

In this manuscript, Craig Brown and colleagues have reported that chemogenetic stimulation of vasoactive intestinal peptide (VIP) interneurons enhances weakened sensory responses in the peri-infarct sensorimotor cortex of mice subjected to photothrombotic stroke and promotes the recovery of sensorimotor function. It is novel that sensory responses were disrupted by stroke

mainly within a population of highly active VIP neurons. The finding that chemogenetic stimulation accelerates the recovery of sensory responses within these highly active VIP neurons and without recruitment of neurons not previously involved in sensory responses is also intriguing. The manuscript is well written; experiments were involving a battery of imaging, electrophysiological and behavioral approaches that require significant expertise. The findings of this paper have clinical significance and support the notion that stroke recovery could be accelerated by enhancing cortical excitability through dis-inhibitory VIP interneurons. Statistical analyses are appropriate, and the level of details provided would be sufficient for other researchers to reproduce the work. The findings will be of interest to the others in the field and the broader readership. I am highly enthusiastic about this manuscript and support its publication.

Addressing several points could further improve this excellent manuscript.

1. Surprisingly, the authors did not come across dying neurons in the peri-infarct cortex during longitudinal imaging. It seems that all VIP interneurons imaged before stroke persisted for the next four weeks of 2P imaging. These findings should be discussed.

2. The peri-infarct zone in the photothrombotic stroke is relatively narrow. All 2P imaging studies were conducted 400 micrometers from the infarct border. Laser speckle images were collected. What percent of the pre-stroke blood flow was remaining in the area of 2P imaging? That could be estimated from the laser speckle images. Were these 2P images collected in the hypo-perfused cortex?

3. The infarct size is large, and spontaneous spreading depolarizations often accompany such large strokes. However, the authors did not record any large increases in the calcium signals associated with spreading depolarizations. It is not surprising since spontaneous changes in GCaMP6s fluorescence were recorded only for periods of up to 75 s after stroke, and there were no longitudinal electrophysiological recordings. It might be a missed opportunity because one possibility is that spreading depolarizations in the penumbra were the events that affected ("wreak havoc on") the predictability (fidelity) of neuronal responses to sensory stimulation (Fig. 6). Certainly, experiments addressing this possibility are beyond the experiments of this study, but such a possibility could be discussed.

1. I like the discussion that "stroke related plasticity usually involves the same circuits"(P.14L31). It might be helpful to argue that even during more severe brain injury inflicted by global ischemia, the postsynaptic dendritic membranes remain attached to axonal boutons, providing a structural basis for the recovery of the same circuits (doi.org/10.1093/cercor/bhaa134).

Minor points:

1. What was the rationale to start CNO treatment on day four but not earlier? Also, the additional explanation on P.15 L9 "(ie. Days 4 to 6, Monday through Wednesday, before imaging on day 7)" seems unnecessary.

2. P21.L23. "... time to peak amplitude and half-width (ie. duration) of forelimb-evoked signals in the first 150ms after stimulation were measured with Clampfit 9.0 software (Molecular Devices)." Does it belong to another section, such as "Recording sensory evoked cortical field potentials"?

Sergei A. Kirov, PhD

Reviewer #3:

Remarks to the Author:

In this study by Motaharinia et al the authors report that activating VIP interneurons with Gq DREADDs after stroke to the forelimb (FL) region of somatosensory cortex can restore cortical responses (to FL stimulation) to pre-stroke levels and enhance functional recovery. This is an important study with robust findings that add significantly to our understanding of stroke recovery. It is the only paper to my knowledge that has reported on the activity of VIP neurons in the context of stroke recovery. It is also an elegant study that combines two-photon calcium imaging in vivo, mouse behavior, and DREADDs. These experiments are hard! Finally, the finding that VIP neurons could be a target for restoring circuit function and ameliorating behavioral deficits after stroke is very exciting. I was impressed by the high experimental rigor: they use CNO-only (no hM3Dq) controls and they use appropriate stats. The blood flow control to rule out DREADD effects is great too. The behavior data shows definitive results with internal replication in 2 cohorts. The

longitudinal imaging of VIP neurons over >4 weeks is particularly impressive because they track the same neurons over time. I felt the paper was well written, the figures are easy to understand, and they provide appropriate references (for example of papers describing the known role of VIP cells in disinhibition of pyramidal cells). Overall, I feel the significance of this paper is high and I am enthusiastic about its publication. It is refreshing to see the use of cutting-edge tools to investigate functional circuit changes at the single cell level after stroke. Here is a list of comments/suggestions I hope the authors can address

- I guess they never show that the excitability/firing of VIP neurons in control mice is enhanced by Gq + CNO. They only show indirectly with LFP that stimulation of FL elicits greater responses (from Pyr cells). I wonder if they ever saw differences with calcium imaging in VIP neurons before and 30 min after CNO.
- I was surprised that a single injection of CNO (which has such a short half-life) only 5 days a week was enough to rescue behavior? How do the authors interpret such a profound effect on the network (and behavior!)? Also, do they think the circuit is permanently restored such that, had they looked a few days after stopping CNO injections, the rescue might have persisted?
- They choose to perform photothrombotic strokes that do not completely destroy S1FL (eg Fig 1a). Presumably this is to make sure they still can find FL-responsive VIP neurons in peri-infarct cortex. The authors should make it more explicit in the text that they performed sub-total strokes, because it is likely that they would have never found any FL-responsive VIP neurons in peri-infarct cortex had they completely destroyed S1FL.
- Fig 1c shows that stroke strongly reduces FL stim evoked responses compared to sham. In Fig 1d peak amplitudes are "normalized to baseline", which means right before CNO injection. Could they show a comparison of these baseline responses (average of all the mice for stroke vs sham) to see how strongly stroke affects the FL stim evoked response (beyond the representative trace in Fig 1)
- In Fig 1d, what are the post-hoc individual p values for the effect of CNO in hM3Dq mice that did or did not receive a stroke?
- I like Suppl Fig 1a better than Fig. 1e. They might consider a swap (optional)
- In suppl Fig 2 please list the number of mice in each group; I don't think a t-test is appropriate here
- Fig 2d: why not show the average DF/F for all ROIs combined (1-6)? It seems like the differences between hM3Dq and control mice were not significant (otherwise all the data would be presented in the same figure panel, like a bar graph of the peak response). The text says "forepaw evoked depolarizations in peri-infarct cortex were SIGNIFICANTLY larger in amplitude ... (Fig. 2b-d)" but there are no stats provided in the text or in the figure legend. It's fine if it's only significant for some of the ROIs (e.g., 4 & 5), but this could have meaning too based on their location relative to the infarct.
- In Fig 2b-f, what is the control? Is it hM3Dq but not CNO, or is it GFP + CNO alone? Or are they combined?
- Fig 3a - is the earliest time point -6 weeks or -3 weeks? They should say in the legend what the green/blue contours of maps represent (presumably it's the 75% threshold described in page 19, line 19).
- Page 8, lines 21-24: they should probably show these data as part of suppl fig 3
- Figs 3 & 4: regarding the concern about toxicity of prolonged GCaMP expression, I agree it's reassuring that the gray traces in Fig 4e (sham) are stable. But why is the +4 wk time point missing for sham controls in Fig 4e-f?
- Fig 4b: are these example traces from a vehicle stroke mouse?
- Fig 4d: can they use different colors (or symbols) for the 3 different types of mice in these scatter plots (sham control, stroke veh and stroke Gq)?
- Fig 4e-f: It's fine to show the normalized data but can they also show the raw data for % of responsive neurons and peak amplitudes for the three groups at baseline (text has results of ANOVA but not actual data)? I ask because there does not appear to be a sustained increase in the % of VIP neurons that respond to FL stimulation after stroke, which means that no new neurons are recruited to respond to FL stim after stroke (DREADDs only maintain the original pool).
- The effects on VIP responsivity are mainly seen at 1-wk post-stroke, but behavioral effects are seen 2-7 weeks after stroke...the discussion touches on this but more could said to explain this difference.
- Fig 5b: Did the relative proportion of each of the 3 subtypes change over time (from baseline) in

sham controls and stroke animals? There should be a graph to represent that. Also, shouldn't there be a Chi-square test for all the comparisons and then follow-up 2x2 chi-sq for individual comparisons with post-hoc correction. A better description of the Chi-sq methods would be useful

- Fig 6a: the issue of how predictable (or should it be 'reliable?') neurons are seems important, but this visual representation is a bit hard to follow. Another way to show this would be to plot, for each cell, the % of stimulations it responds to at baseline and over time after stroke.

RESPONSE TO REVIEWER COMMENTS

Dear editors and reviewers:

We would like to thank you for the opportunity to revise our manuscript titled “Longitudinal functional imaging of VIP interneurons reveals sup-population specific effects of stroke that can be rescued with chemogenetic therapy” for publication in *Nature Communications*. The reviewers have provided a number of important and incisive suggestions for improving the paper. As requested by the editors and reviewers, we have conducted several new and challenging experiments to support the conclusions in the paper. These experiments include: a) electrophysiological DC recordings and *in vivo* GCaMP6s imaging of stroke related spreading depolarizations (see Reviewer 2 comments), b) *in vivo* imaging of VIP sensory responses followed by post-mortem re-identification of VIP neurons that co-localize with calretinin (see Reviewer 1 comments) and c) imaging the acute effects of chemogenetic stimulation on VIP neuron responses (see Reviewer 3 comments). In addition, we have included several new data analyses at the request of the reviewers. Please note that all manuscript revisions are highlighted in red text. As a result of these recommendations and revisions, we have strengthened the evidence supporting the paper’s primary findings and refined our discussion of these results.

Reviewer #1 (Remarks to the Author):

Motaharinia et al. explored whether chemogenetic modulation of VIP+ interneurons in the somatosensory cortex of mice can be beneficial for stroke recovery. The authors provide very interesting data with longitudinal imaging of VIP+ interneurons before and several weeks after local stroke induction. The major finding is that driving VIP+ cells chemogenetically can rescue cortical excitability after stroke and fasten functional recovery. The data is of highest quality and were obtained in carefully designed experiments. I only have several minor comments/questions to clarify some findings.

RESPONSE: We thank the reviewer for their positive appraisal of our study.

1. The manuscript gives an impression that the major role of neocortical VIP+ cells is disinhibitory; however, this may not be the case. VIP+ neuronal population is highly heterogeneous, with up to 80% of VIP+ contacts formed onto GABA negative structures (Zhou et al., 2017). This heterogeneity may result in highly variable patterns of activity observed by the authors in VIP+ cells, and needs to be carefully discussed. For example, across cortical areas, VIP+/CCK+ basket cells contact PYRs, whereas VIP+/CR+ cells contact preferentially interneurons (Guet-McCreight et al., 2020). Importantly, these two VIP+ subtypes have different intrinsic excitability, with VIP+/CR+ cells showing much higher input resistance. Accordingly, the disinhibitory VIP+/CR+ cells may represent the highly active group in this study and a preferential target for post-stroke therapy. It seems the authors are able to find VIP+ cells imaged *in vivo* for post-mortem IHC identification (Fig. 3B). It would be therefore possible to check whether the highly active group is expressing CR.

RESPONSE: This is an interesting point, and it would certainly be fascinating to further define the neurochemical phenotype of the highly active VIP neurons. While we fully agree with the reviewer's comment about functional/neurochemical heterogeneity in the VIP population, our claim of a disinhibitory effect, at least on the network/population level, was based on our field recording data showing an increase in cortical field potential responses following DREADD based stimulation of VIP neurons. The reviewer raises the possibility that the highly active VIP neurons may express Calretinin (CR), which show much higher input resistance. Therefore to address this possibility, we installed cranial windows and longitudinally imaged 10 mice to assess VIP sensory responses, then attempted to find each individual neuron in post-mortem horizontal sections immunostained for Calretinin (CR). Since these are extremely difficult experiments with very low success rates, we could only re-locate the VIP neurons imaged *in vivo* in 2 mice. In these mice, we found that 34% VIP neurons expressing GCaMP6s co-localized with CR (see **Supp. Fig. 6a,b**). Conversely, 31.5% of CR neurons co-localized with VIP. Since there were few VIP neurons that co-localized with CR to sample from, we graphed the number of responsive trials (out of 8) as function of whether VIP neurons co-localized with CR or not (2 mice: 71 neurons were VIP+/CR- and 14 neurons were VIP+/CR+). As shown in **Supp. Fig. 6c**, the response profile of VIP neurons that co-express calretinin versus those that do not, were not clearly different from one another. Based on these results, we conclude that the highly responsive group of VIP neurons (ie. neurons that respond to 6-8 trials) are not exclusively the same ones that express CR. However, we agree with the reviewer that future studies probing the response profile of VIP neurons with specific neuropeptides or calcium binding proteins (CR, CCK, ChaT etc) would be informative. We have included this topic in the discussion on page 18 and presented the results of this experiment in a new **Supp. Fig. 6**.

2. Was there any difference in evoked field response for 0.3 vs. 0.5 mg/kg of CNO? Why not using the same dose in all experiments? Ideally, one would use the lowest efficient dose throughout the entire study.

RESPONSE: Ideally, yes we would have used the same dose. Since the CNO-DREADD experiments were carried out in 2 separate epochs (2013-2015 and 2017-present), there was a mis-communication when we re-started the CNO experiments in the second epoch (trainee thought we were using 0.5mg/kg vs 0.3mg/kg). However in both epochs, we tested and validated the CNO-DREADD effects with electrophysiology to ensure they would work as expected. We now present our analysis of peak field potential responses when mice were dosed with 0.3 vs. 0.5mg/kg CNO. Importantly we show there was no significant differences between doses and now include this analysis in the results on page 5 "There was no significant difference in peak response amplitude when comparing 0.3mg/kg vs 0.5mg/kg doses of CNO (unpaired t-test, $t_{(3)}=1.21$, $p=0.31$)."

3. Why the therapy had started 4 days after stroke? Would a faster intervention be more beneficial? Have the authors tried different time intervals after the stroke, like 1, 2, 3 days?

RESPONSE: In the first 72 hours after stroke, the brain is in quite a labile and precarious state due to edema, spreading depolarizations, excitotoxicity and other potentially damaging events that can further expand the area of ischemic damage. Since our goals were to: a) focus on stroke recovery strategies rather than neuroprotection (ie. enhance function of what brain tissue remains vs. preventing ischemic cell death), and b) image the same "peri-infarct" neurons before and after stroke; we did not want to further risk aggravating ischemic cell death by chemogenetically augmenting excitability within the first 3 days after stroke. We have now included a statement in the results clarifying our rationale (page 6). Lastly, we

do agree that an interesting future study would be to test out different time points for initiating therapeutic intervention (Note: this actually part of a recent grant proposal), especially if we could extent the therapeutic window of opportunity.

4. Were there any acute affects of CNO treatment?

RESPONSE: As shown in Figure 1e and Supp. Fig. 1, the effects of CNO treatment on cortical excitability peak within the first 60-90min after injection and then, according to published literature (Alexander et al., 2009, Neuron), start to decline thereafter.

5. Was the beneficial effect of chemogenetic treatment persistent? Or have the authors evaluated the state of animals at a longer scale – 3-4 weeks (up to a month) after therapy cessation?

RESPONSE: We too were curious about this. Our behavioural data suggest that the beneficial effects did persist given that we stopped treatment at week 6, and then even one week later, the mice that received chemogenetic treatment were still significantly better than controls on the horizontal ladder test (see Fig. 2a). Furthermore, when examining forelimb evoked cortical responses with VSD imaging at 10 weeks recovery (~ 4 weeks after treatment was stopped), the mice that received chemogenetic treatment showed significantly increased sensory-evoked cortical responses compared to controls (see Fig. 2b-e).

6. It is exciting to see that a single dose of CNO, with VIP activation time course of about 2 h (max up to 10 h), produces such a long-lasting (at least 23 h) effect on cortical excitability. Can you discuss a bit more the possible mechanisms of this plasticity?

RESPONSE: This is a good point. The fact that the effects of chemogenetic therapy persist long after treatment has ceased suggest that up-regulating the excitability of VIP neurons has led to more permanent changes within stroke recovered circuits, presumably those downstream of VIP neurons (eg. layer 2/3 pyramidal neurons and other interneurons). Precisely how these lasting downstream changes in circuitry are accomplished remains an open question. Previous work from our lab and others has shown that promoting the restoration of cortical excitability in somatosensory cortex after stroke (via chemo- or optogenetics) is associated axonal sprouting (Wahl et al., 2017, Nature Comm) as well as the proliferation and stabilization of thalamocortical axonal boutons (Tennant et al., 2017, Nature Comm). In addition, therapies that promote the return of cortical excitability after stroke lead to changes in growth and plasticity associated gene expression (CREB, BDNF, NGF; see Cheng et al., 2014, PNAS; Caracciolo et al., 2018, Nat Comm). Although our study provides much needed insights into the effects of stroke at a cellular level, future stroke studies could dissect the contribution of other interneuron populations in stroke recovery. Furthermore, we think the stroke field could really benefit from future slice electrophysiology studies that address circuit specific changes in intrinsic excitability, spiking patterns, pre and post-synaptic forms of plasticity (LTP, LTD) using quantal analysis, paired pulse ratios, mini-analysis etc. We now include a more fulsome discussion on this topic on page 14 of the discussion.

7. In Discussion, the authors propose that chemogenetic manipulation can represent a relatively non-invasive way for boosting excitability. I would disagree with this point, as it requires stereotaxic surgery to deliver the viral vectors expressing the DREADD. The obvious advantage of this method is, however, its “targetability” or a possibility of focal manipulations with a local neuronal circuits in peri-infarct area.

RESPONSE: we agree and have removed the phrase about being “non-invasive”

8. Given that male and female mice were used in the study, were there any sex-specific differences in VIP+ cell activity or other reported phenomena?

RESPONSE: For our study, we used male VIP mice for the electrophysiology, VSD and calcium imaging experiments, as well as the first cohort of behavioural studies testing the chemogenetic therapy (see cohort 1, K.G. 2014 in Figure 2a). However as we (and science in general) have become more cognizant of considering sex in our studies, we used female VIP mice for the second cohort of behavioural studies testing the chemogenetic therapy (see Cohort 2, S.C. 2018 in Figure 2a). If we compare males and females that received the chemogenetic therapy relative to their respective control groups, the benefits of therapy for males and females (ie. comparing difference in % correct steps between treated and controls from weeks 2-7), were quite similar and not significantly different (2-way ANOVA; Main effect of Sex: $F_{(1,72)}=1.31$, $p=0.26$). We have now included this analysis in the results (page 6-7) and clarified the sex of the mice in the methods section and Figure legend 2.

Reviewer #2 (Remarks to the Author):

In this manuscript, Craig Brown and colleagues have reported that chemogenetic stimulation of vasoactive intestinal peptide (VIP) interneurons enhances weakened sensory responses in the peri-infarct sensorimotor cortex of mice subjected to photothrombotic stroke and promotes the recovery of sensorimotor function. It is novel that sensory responses were disrupted by stroke mainly within a population of highly active VIP neurons. The finding that chemogenetic stimulation accelerates the recovery of sensory responses within these highly active VIP neurons and without recruitment of neurons not previously involved in sensory responses is also intriguing. The manuscript is well written; experiments were involving a battery of imaging, electrophysiological and behavioral approaches that require significant expertise. The findings of this paper have clinical significance and support the notion that stroke recovery could be accelerated by enhancing cortical excitability through dis-inhibitory VIP interneurons. Statistical analyses are appropriate, and the level of details provided would be sufficient for other researchers to reproduce the work. The findings will be of interest to the others in the field and the broader readership. I am highly enthusiastic about this manuscript and support its publication.

RESPONSE: We thank the reviewer for their positive comments on our study.

Addressing several points could further improve this excellent manuscript.

1. Surprisingly, the authors did not come across dying neurons in the peri-infarct cortex during longitudinal imaging. It seems that all VIP interneurons imaged before stroke persisted for the next four weeks of 2P imaging. These findings should be discussed.

RESPONSE: We think this might be an issue of semantics, but it is an important one. Since we could not control the extent of ischemic damage with micron precision, some areas we had imaged before stroke and had hoped would remain after the induction of stroke, were ultimately destroyed. These areas we considered part of the infarct core, not peri-infarct cortex. Thus, when we imaged neurons 7 days after stroke, we only focused on ones that were viable and outside of the infarct core, which we define as “peri-infarct”. As mentioned below, the infarct border is quite sharp and therefore ischemic cell death was not still evolving by post-stroke day 7. In this context, it is not surprising that neurons imaged 7 days after

stroke, would persist for the remaining 4 weeks. We have clarified this point on page 25 in the methods section.

2. The peri-infarct zone in the photothrombotic stroke is relatively narrow. All 2P imaging studies were conducted 400 micrometers from the infarct border. Laser speckle images were collected. What percent of the pre-stroke blood flow was remaining in the area of 2P imaging? That could be estimated from the laser speckle images. Were these 2P images collected in the hypo-perfused cortex?

RESPONSE: In the present study, we used laser speckle imaging to define the infarct border starting 7 days after stroke, but not before stroke. Therefore, a comparison of pre vs post-stroke blood flow was not possible. However, there are several published papers including our own (Tennant et al., 2013; Sullender, et al., 2018) that show relatively normal blood flow in peri-infarct cortex (after photothrombotic stroke) by 5-7 days post-stroke. We now include this information and cited literature on page 25 in the methods section.

3. The infarct size is large, and spontaneous spreading depolarizations often accompany such large strokes. However, the authors did not record any large increases in the calcium signals associated with spreading depolarizations. It is not surprising since spontaneous changes in GCaMP6s fluorescence were recorded only for periods of up to 75 s after stroke, and there were no longitudinal electrophysiological recordings. It might be a missed opportunity because one possibility is that spreading depolarizations in the penumbra were the events that affected (“wreak havoc on”) the predictability (fidelity) of neuronal responses to sensory stimulation (Fig. 6). Certainly, experiments addressing this possibility are beyond the experiments of this study, but such a possibility could be discussed.

RESPONSE: The idea that ischemic spreading depolarization (SD) could have been occurring 7 days after stroke and thereby influence our findings is an interesting one and something we had not previously considered. Before we describe our new experiments and analyses, we should prevent any confusion by clarifying that all calcium imaging in the original submission was conducted 7 days after stroke, not in the first few minutes to hours after stroke. We apologize for any confusion and have now made this clear in the results and methods section. In order to address the possibility of SDs contaminating our calcium imaging data (collected at 7 days post-stroke), we employed two different, but complimentary approaches (n=6 mice for each experimental approach). First, we used the gold standard approach of recording direct current (DC) potentials in peri-infarct cortex immediately after stroke and 7 days recovery (n=3 mice per time point). Recording immediately after initiating photothrombosis revealed approximately 1-2 SD's occur within the first 30 minutes after photothrombotic stroke (2 mice exhibited 2 SD each whereas 1 mouse showed just 1 SD; peak amplitude: 12 ± 1.1 mV; duration: 161.8 ± 5.2 s; **Supp. Fig. 5a**), which is consistent with previous studies (Risher et al., JNsci, 2010). However, we did not observe any SDs in peri-infarct cortex 7 days after stroke (**Supp Fig. 5a**). To be extra careful, we also included a positive control experiment with 1mM KCl application to prove that if a SD had occurred, we would have detected it. Similarly, by imaging SD induced calcium waves in peri-infarct cortex (which are ~10 times larger in amplitude than sensory evoked calcium transients), we typically observed an SD within the first 30min after stroke (see example shown in **Supp. Fig. 5b**), but never 7 days post-stroke (n=3 mice per time point). Based on these experiments and previous literature, we conclude that it is very unlikely that ongoing SD waves could have affected our calcium imaging experiments and analysis collected 7 days after stroke. We now include these results on page 10 and Supplementary Fig. 5, as well as provided methodological details on page 21 and 25 in the Methods section.

4. I like the discussion that “stroke related plasticity usually involves the same circuits”(P.14L31). It might be helpful to argue that even during more severe brain injury inflicted by global ischemia, the postsynaptic dendritic membranes remain attached to axonal boutons, providing a structural basis for the recovery of the same circuits (doi.org/10.1093/cercor/bhaa134).

RESPONSE: We thank the reviewer for this suggestion and agree that if post-synaptic structures retain their pre-synaptic partners, this could provide a structural explanation for the temporary disruption and re-activation of highly active circuits. We now include this relevant paper and explanation in the discussion section on page 16.

Minor points:

1. What was the rationale to start CNO treatment on day four but not earlier? Also, the additional explanation on P.15 L9 “(ie. Days 4 to 6, Monday through Wednesday, before imaging on day 7)” seems unnecessary.

RESPONSE: We have removed the unnecessary information. Since Reviewer 1 asked a similar question about the timing of treatment (query 3), we have copied part of our response here.

“In the first 72 hours after stroke, the brain is in quite a labile and precarious state due to edema, spreading depolarizations, excitotoxicity and other potentially damaging events that can further expand the area of ischemic damage. Since our goals were to: a) focus on stroke recovery strategies vs. neuroprotection (ie. enhance function of what brain tissue remains vs. preventing ischemic cell death), and b) image the same “peri-infarct” neurons before and after stroke; we did not want to further risk aggravating ischemic cell death by chemogenetically augmenting excitability within the first 3 days after stroke. We have now included a statement in the results clarifying our rationale (page 6).”

2. P21.L23. “... time to peak amplitude and half-width (ie. duration) of forelimb-evoked signals in the first 150ms after stimulation were measured with Clampfit 9.0 software (Molecular Devices).” Does it belong to another section, such as “Recording sensory evoked cortical field potentials”?

RESPONSE: This is correct, we analysed VSD signals in this manner.

Reviewer #3 (Remarks to the Author):

In this study by Motaharinia et al the authors report that activating VIP interneurons with Gq DREADDs after stroke to the forelimb (FL) region of somatosensory cortex can restore cortical responses (to FL stimulation) to pre-stroke levels and enhance functional recovery. This is an important study with robust findings that add significantly to our understanding of stroke recovery. It is the only paper to my knowledge that has reported on the activity of VIP neurons in the context of stroke recovery. It is also an elegant study that combines two-photon calcium imaging in vivo, mouse behavior, and DREADDs. These experiments are hard! Finally, the finding that VIP neurons could be a target for restoring circuit function and ameliorating behavioral deficits after stroke is very exciting.

I was impressed by the high experimental rigor: they use CNO-only (no hM3Dq) controls and they use appropriate stats. The blood flow control to rule out DREADD effects is great too. The behavior data shows definitive results with internal replication in 2 cohorts. The longitudinal imaging of VIP neurons

over >4 weeks is particularly impressive because they track the same neurons over time. I felt the paper was well written, the figures are easy to understand, and they provide appropriate references (for example of papers describing the known role of VIP cells in disinhibition of pyramidal cells). Overall, I feel the significance of this paper is high and I am enthusiastic about its publication. It is refreshing to see the use of cutting-edge tools to investigate functional circuit changes at the single cell level after stroke. Here is a list of comments/suggestions I hope the authors can address

RESPONSE: We thank the reviewer for their encouraging comments on our study.

1. I guess they never show that the excitability/firing of VIP neurons in control mice is enhanced by Gq + CNO. They only show indirectly with LFP that stimulation of FL elicits greater responses (from Pyr cells). I wonder if they ever saw differences with calcium imaging in VIP neurons before and 30 min after CNO.

RESPONSE: Since there have been dozens of papers published validating the excitatory effects of hM3Dq in neurons, our primary question was whether stimulating VIP neurons would lead to a general increase in sensory evoked cortical responses. This was the first set of experiments we conducted and provided the rationale for doing the entire study. Originally in 2013-14, the hM3Dq AAV2 was provided by Penn state core vector facility and my post-doc K.G. validated the idea that chemogenetic stimulation of VIP neurons would enhance sensory evoked responses in both sham control and stroke mice. K.G. left for maternity leave (and ultimately science in 2016), when M.M. continued the project in 2017-present. However the hM3Dq AAV was now only provided by Addgene, so we again did electrophysiology experiments to prove the hM3Dq worked as expected. We should note that there was no significant difference in peak response amplitude (following 0.3 or 0.5mg/kg CNO) between these two sets of validation experiments (this info is now provided on page 5 of the results). However since our sample was small for validating the effects of hM3Dq in stroke mice (see our response to query 5 below), we had 3 VIP-cre mice left in our colony that had been injected with cre-dependent hM3Dq + GCaMP6s and had recovered for ~14 weeks from stroke in FLS1 cortex. Although imaging conditions were not ideal (this is expected many months after implantation of a cranial window, especially if a stroke is involved), we found some forelimb responsive VIP neurons and imaged their responses before and after i.p. injection of 0.5mg/kg CNO. These results show a significant increase in mean forelimb evoked response amplitude and # responsive trials after CNO injection and are now presented in Supp. Fig. 1f. Thus, we believe that we have validated the use of chemogenetic treatment with multiple experiments from different experimenters at different epochs in time, using different but complimentary approaches (electrophysiology, calcium/VSD imaging, behaviour).

2. I was surprised that a single injection of CNO (which has such a short half-life) only 5 days a week was enough to rescue behavior? How do the authors interpret such a profound effect on the network (and behavior!)? Also, do they think the circuit is permanently restored such that, had they looked a few days after stopping CNO injections, the rescue might have persisted?

RESPONSE: This is a good point for the discussion section. Our cortical field recording experiments indicated that the effect of CNO on network excitability lasted for at least 60 min in our VIP mice (Fig. 1 and Supp. Fig. 1), and could potentially last longer based on the work of other groups (Alexander et al., 2009, Neuron). The idea that 60 minutes of chemogenetic therapy 5 days a week can positively affect behavioural outcome is in line with our previous paper (see Tennant et al; 2017, Nature Comm) where we

provided intermittent optogenetic stimulation for 1 hour/day for 5 days a week for up to 6 weeks. Our present experiments indicate the benefits of chemogenetic therapy persist long after treatment has ceased and therefore likely involve permanent changes in circuitry. For example, the improvement in ladder walking performance is maintained at week 7 in the stimulated group even though treatment had ceased at week 6. Similarly, VSD imaging was conducted at week 10 (4 weeks after treatment had ceased) and we still see an enhancement in sensory evoked response amplitudes in the stimulated group. Given that layer 2/3 pyramidal neurons are the dominant source of signal for VSD imaging, our findings imply these layer 2/3 circuits were strengthened with chemogenetic therapy. Precisely how these lasting downstream changes in circuitry are accomplished remains an open question. Previous work from our lab and others has shown that promoting the restoration of cortical excitability in somatosensory cortex after stroke (via chemo- or opto-genetics) is associated axonal sprouting (Wahl et al., 2017, Nature Comm) as well as the proliferation and stabilization of thalamocortical axonal boutons (Tennant et al., 2017, Nature Comm). In addition, therapies that promote the return of cortical excitability after stroke lead to changes in growth or plasticity associated gene expression (CREB, BDNF, NGF; see Cheng et al., 2014, PNAS; Caracciolo et al., 2018, Nat Comm). Although our study provides much needed insights into the cellular effects of stroke, at least within VIP interneurons, future stroke studies could dissect the contribution of other interneuron populations in stroke recovery. Furthermore, we think the stroke field could really benefit from future brain slice electrophysiology studies that address circuit specific changes in intrinsic excitability, spiking patterns, pre and post-synaptic forms of plasticity (LTP, LTD) using quantal analysis, paired pulse ratios, mini-analysis etc. We now include a more fulsome discussion on this topic on page 14 of the discussion.

3. They choose to perform photothrombotic strokes that do not completely destroy S1FL (eg Fig 1a). Presumably this is to make sure they still can find FL-responsive VIP neurons in peri-infarct cortex. The authors should make it more explicit in the text that they performed sub-total strokes, because it is likely that they would have never found any FL-responsive VIP neurons in peri-infarct cortex had they completely destroyed S1FL.

RESPONSE: The reviewer is correct that we did not destroy the entire FL cortex or else there would be little or no forelimb responsive neurons to image. We have now made this clear on page 5 of the results section.

4. Fig 1c shows that stroke strongly reduces FL stim evoked responses compared to sham. In Fig 1d peak amplitudes are “normalized to baseline”, which means right before CNO injection. Could they show a comparison of these baseline responses (average of all the mice for stroke vs sham) to see how strongly stroke affects the FL stim evoked response (beyond the representative trace in Fig 1)

RESPONSE: In accordance with the reviewer’s suggestion, we have added the average forelimb-evoked responses at baseline and following CNO injection for both stroke and sham stroke mice. This data is now presented in Supp. Fig. 1b.

5. In Fig 1d, what are the post-hoc individual p values for the effect of CNO in hM3Dq mice that did or did not receive a stroke?

RESPONSE: As discussed in response to query 1, the excitatory effects of the hM3Dq were validated twice with electrophysiology by two different experimenters at two different points in time (K.G.

validated in 2013-14 and M.M. validated in 2017-18). We combined these two validation experiments together (n=7 mice) to show a very robust excitatory effect of CNO on cortical sensory responses with post-hoc p values (see Fig. 1d). We have also added a new figure showing peak forelimb evoked field amplitudes before vs after CNO injection for individual mice in each of the experimental conditions (see Supp. Fig. 1a). However, the 7d stroke data was from a small sample. Although both stroke mice expressing hM3Dq showed a very clear increase in forelimb responses 30min after CNO injection relative to before (see new Supp. Fig. 1a), concerns about sampling should be mollified by the addition of new calcium imaging data from stroke recovered mice showing an increase in response amplitude and # responsive trials following CNO treatment (see our response to query 1, data shown in Supp. Fig. 1f). Therefore, despite the small sample for that specific experiment, there are multiple converging lines of experimental evidence showing that the chemogenetic treatment increases cortical responses to sensory stimulation in sham stroke controls or stroke recovered mice.

6. I like Suppl Fig 1a better than Fig. 1e. They might consider a swap (optional)

RESPONSE: As requested by the reviewer, we have swapped these figures.

7. In suppl Fig 2 please list the number of mice in each group; I don't think a t-test is appropriate here

RESPONSE: We have now added the number of mice in each group to Supp. Fig. 2 legend

8. Fig 2d: why not show the average DF/F for all ROIs combined (1-6)? It seems like the differences between hM3Dq and control mice were not significant (otherwise all the data would be presented in the same figure panel, like a bar graph of the peak response). The text says “forepaw evoked depolarizations in peri-infarct cortex were SIGNIFICANTLY larger in amplitude ... (Fig. 2b-d)” but there are no stats provided in the text or in the figure legend. It's fine if it's only significant for some of the ROIs (e.g., 4 & 5), but this could have meaning too based on their location relative to the infarct.

RESPONSE: We agree and have now added the average forelimb evoked dF/Fo (shaded area represents SEM) for all ROIs and all mice within each group (see new traces added to Fig. 2d). The reviewer also makes a good point about the statistical description. To clarify, the omnibus statistical analysis of forelimb-evoked responses (2-way ANOVA) was conducted only on normalized data to control for considerable between-experiment variability in response amplitudes. This analysis showed a significant effect of CNO treatment which is reported in figure 2e and the accompanying legend. We have removed the word “significant” for the statement pertaining to un-normalized data and reserved it for the analysis of normalized data in Fig. 2e.

9. In Fig 2b-f, what is the control? Is it hM3Dq but not CNO, or is it GFP + CNO alone? Or are they combined?

RESPONSE: The stroke control group for VSD experiments was run by author K.G and consisted of mice that expressed hM3Dq, but received vehicle injections instead of CNO. We have now clarified this in the figure 2 legend.

10. Fig 3a - is the earliest time point -6 weeks or -3 weeks? They should say in the legend what the

green/blue contours of maps represent (presumably it's the 75% threshold described in page 19, line 19).

RESPONSE: -6 weeks (before stroke) is when the AAV injection and cranial window were installed, whereas imaging began 4 weeks later (-2 weeks before stroke). The reviewer is correct in that green/blue contours were derived from a 75% of peak amplitude threshold. We now clarify this in the Figure 3 legend.

11. Page 8, lines 21-24: they should probably show these data as part of suppl fig 3

RESPONSE: As requested, we have inserted a new graph in Supp Fig 3a-c showing baseline % responsive neurons, % responsive trials and peak amplitudes for the 3 groups.

12. Figs 3 & 4: regarding the concern about toxicity of prolonged GCaMP expression, I agree it's reassuring that the gray traces in Fig 4e (sham) are stable. But why is the +4 wk time point missing for sham controls in Fig 4e-f?

RESPONSE: The sham stroke control mice (grey traces) were imaged for 4 weeks (BL and 3 weeks after sham stroke). Since neural responses were quite stable over this period of time, we did not collect data for the +4 week time point.

13. Fig 4b: are these example traces from a vehicle stroke mouse?

RESPONSE: The reviewer is correct as the traces are from the same cells shown in Fig. 4a. We have clarified this point in the figure legend.

14. Fig 4d: can they use different colors (or symbols) for the 3 different types of mice in these scatter plots (sham control, stroke veh and stroke Gq)?

RESPONSE: The scatterplot only shows data from stroke-affected mice that received control treatment since sham stroke or CNO treated stroke mice did not exhibit a loss of sensory responses at 1 week after stroke.

15. Fig 4e-f: It's fine to show the normalized data but can they also show the raw data for % of responsive neurons and peak amplitudes for the three groups at baseline (text has results of ANOVA but not actual data)? I ask because there does not appear to be a sustained increase in the % of VIP neurons that respond to FL stimulation after stroke, which means that no new neurons are recruited to respond to FL stim after stroke (DREADDs only maintain the original pool).

RESPONSE: As requested by the reviewer, we have now plotted this raw baseline data in Supp. Fig. 3a-c. The reviewer is correct in that there wasn't a significant or sustained increase in the % responsive VIP neurons after stroke in CNO treated mice relative to pre-stroke. As shown in Fig. 4e, if this were true we would have denoted the 1 week data with a # symbol, which represents statistical comparisons of post-stroke values vs. pre-stroke. We agree with the reviewer's conclusion that new forelimb neurons are not recruited after stroke and had previously stated this in the abstract and discussion. We have now included

the following conclusion statement in the results on page 12 to make this point clear: “Further, our data argue against the idea that new forelimb responsive neurons are recruited after stroke.”

16. The effects on VIP responsivity are mainly seen at 1-wk post-stroke, but behavioral effects are seen 2-7 weeks after stroke...the discussion touches on this but more could said to explain this difference.

RESPONSE: We agree that this is an interesting finding but at this point in time, we can only speculate why the behavioural effects lag by 1 week behind the VIP response differences. It could be that the post-synaptic targets of VIP neurons, or ones further downstream (eg. pyramidal neurons) are slightly slower to restore intrinsic excitability or synaptic drive. As mentioned by reviewer 1, we think future studies imaging the downstream targets of VIP neurons or whole cell recordings in brain slices in defined excitatory and inhibitory circuits would help resolve this question (see new text added to page 14 of the discussion). However, these experiments would require years of work and therefore are best reserved for future study.

17. Fig 5b: Did the relative proportion of each of the 3 subtypes change over time (from baseline) in sham controls and stroke animals? There should be a graph to represent that. Also, shouldn't there be a Chi-square test for all the comparisons and then follow-up 2x2 chi-sq for individual comparisons with post-hoc correction. A better description of the Chi-sq methods would be useful

RESPONSE: Although time dependent changes in the relative proportion of each of the 3 subtypes is depicted and analysed in Fig. 5b, we agree with the reviewer that an additional graph (perhaps simplified) showing the proportion of each subtype within each experimental group, would be informative. Therefore we have added a new graph into Figure 5 (see new Fig. 5d) that shows the % of each subtype before stroke compared to after stroke (values for weeks 1-3 averaged together, similar to Fig. 5b) in each of the 3 experimental groups. We now report on page 11-12 of the results that the proportion of each subtype does not change significantly in the sham stroke controls or stroke affected mice that received chemogenetic stimulation (Note: this data further supports the idea that chemogenetic stimulation was beneficial). This conclusion was based on a chi-squared analysis where we used pre-stroke as the “expected” proportion for each neuron subtype and then compared that to the “observed” proportion averaged over 3 weeks after stroke (all χ^2 values ranged from 0.01-0.38 and therefore none were close to significant). By contrast, stroke mice that received control stimulation exhibited significantly fewer highly active neurons ($\chi^2 = 4.91$, * $p < 0.05$) and more minimally active neurons after stroke ($\chi^2 = 13.73$, ** $p < 0.01$). This finding is consistent with the idea that stroke has a dampening effect on sensory responses. Since we compared the % cells within each experimental group (or % neuron sub-type) and analysed the average values over 3 weeks post-stroke rather than compare each week, we believe the direct statistical comparison to Sham stroke or Pre-stroke (as shown in Fig. 5b and Fig. 5d, respectively) was justified. We have added further clarity in the statistics section of the Methods (page 27) regarding how we calculated the chi-square statistic.

18. Fig 6a: the issue of how predictable (or should it be ‘reliable’?) neurons are seems important, but this visual representation is a bit hard to follow. Another way to show this would be to plot, for each cell, the % of stimulations it responds to at baseline and over time after stroke.

RESPONSE: We agree this result was not the easiest to present or describe, but think it is quite important. We tried many different graphs/approaches and asked our colleagues what they thought was

easiest/simplest to interpret. The graph presented in Fig. 6a was the “winning” graph and therefore we wish to keep it as presented.

Reviewers' Comments:

Reviewer #1:

Remarks to the Author:

The author have adequately addressed all critical points and clarified their observations according to the comments provided. The manuscript is largely improved.

Reviewer #2:

Remarks to the Author:

The authors have addressed all my previous questions. This is an excellent manuscript and I support publication of this interesting work.

Sergei A Kirov

Reviewer #3:

Remarks to the Author:

The authors have meticulously responded to all of the concerns and comments I had raised and I have no further issues. I congratulate the authors on a beautiful and compelling study.

REVIEWERS' COMMENTS

Reviewer #1 (Remarks to the Author):

The author have adequately addressed all critical points and clarified their observations according to the comments provided. The manuscript is largely improved.

Reviewer #2 (Remarks to the Author):

The authors have addressed all my previous questions. This is an excellent manuscript and I support publication of this interesting work.

Sergei A Kirov

Reviewer #3 (Remarks to the Author):

The authors have meticulously responded to all of the concerns and comments I had raised and I have no further issues. I congratulate the authors on a beautiful and compelling study.